# ON THE STABILITY OF NONLINEAR DYNAMICS IN GD AND SGD: BEYOND QUADRATIC POTENTIALS

## ABSTRACT

The dynamical stability of the iterates during training plays a key role in determining the minima obtained by training algorithms. For example, stable solutions of gradient descent (GD) correspond to flat minima, and these have been associated with favorable features. While prior work often relies on linearization to determine stability, it remains unclear whether linearized dynamics faithfully capture the full nonlinear behavior. In this work, we explicitly study the effect of nonlinear terms. For GD, we show that linear analysis can be misleading. The iterates may stably oscillate near a linearly unstable minimum, and still converge once the step size decays. Here, we derive an exact condition for such stable oscillations, which depends on higher-order derivatives of the loss. Extending the analysis to stochastic gradient descent (SGD), we demonstrate that nonlinear dynamics can diverge in expectation if even a single batch is unstable. This implies that stability can be dictated by the worst-case batch, rather than an average effect, as linear analysis suggests. Finally, we prove that if all batches are linearly stable, then the nonlinear dynamics of SGD are stable in expectation.

## 1 INTRODUCTION

Understanding the nature of the minima reached by our training procedures is a central problem in machine learning and optimization (Neyshabur et al., 2014). A common way to investigate this issue is by analyzing the stability of the iterates as the algorithm approaches a minimum (Wu et al., 2018). For example, it has been shown that stable minimizers of gradient descent (GD) correspond to flat minima (Cohen et al., 2021), and those have been associated with flat predictor functions (Mulayoff et al., 2021; Nacson et al., 2023) and balanced networks (Mulayoff & Michaeli, 2020). These highlight the role of dynamical stability in shaping the properties of the obtained solutions.

In dynamical systems theory, stability analysis is often carried out via linearization. Once the iterates arrive at the vicinity of a fixed point, it is often sufficient to study the linearized system in order to determine whether convergence occurs (Thompson & Stewart, 2002). This technique has been widely applied to study GD (Cohen et al., 2021). Extending it to the stochastic regime, Wu et al. (2018) proposed using linearization to analyze the stability of stochastic gradient descent (SGD) in the mean-square sense.

This approach has inspired a large body of subsequent research. In particular, Ma & Ying (2021) demonstrated that the moments of the linearized dynamics evolve independently, and for the second moment (mean squared error), they provided an implicit expression for the exact stability criterion. Building on this result, Mulayoff & Michaeli (2024) derived an explicit form of the condition, yielding new insights into the linear stability of SGD. Importantly, the stability threshold of the step size depends on the curvature of all samples in the training set (see App. B). Despite this progress, however, it remains unclear whether—and under what conditions—the behavior of linearized iterates truly reflects the full nonlinear dynamics of SGD.

In this work, we aim to address this gap by studying the effect of nonlinear terms. We begin with the deterministic case of GD, as we observe that linearized dynamics can be misleading. GD's iterates may stably oscillate near a linearly unstable minimum and, after step size decay, eventually converge to it. This indicates that oscillations must be taken into account while considering minima stability. Such oscillations correspond to a flip (period doubling) bifurcation of the GD map, in which the

iterates oscillate along the sharpest direction of the minimum. The stability of this bifurcation is governed by the first Lyapunov coefficient in its normal form (see Sec. 5.1). Using this understanding, we derive a precise criterion for stable oscillations. This condition depends on the third- and fourth-order derivatives of the loss at the minimum (see Thm. 1).

We then extend our analysis to the stochastic setting of SGD. Following prior work, we focus on interpolating minima and assume the loss functions are analytic in a neighborhood of the minimum. In this setting, linearized dynamics, combined with mean-square analysis, suggest that the stability threshold of SGD depends on an average curvature over the different mini-batches. In contrast, we show that if the iterates are unstable even with respect to a single batch, the full nonlinear dynamics of SGD are unstable in expectation. This result suggests that stability is determined by the worst-case batch, contradicting prior assumptions about the averaging effect of stochasticity (see Thm. 2).

Finally, we provide a sufficient condition for the stability of SGD. Specifically, we prove that if the dynamics are linearly stable with respect to all possible batches, then there exists a neighborhood of the minimum from which the full nonlinear dynamics converge in expectation (see Thm. 3). Our analysis uses Koopman theory (Koopman, 1931), which allows us to formulate the finite-dimensional nonlinear dynamics as a linear dynamical system in an infinite-dimensional Hilbert space. This reformulation yields two notable benefits. First, nonlinear dynamics are reduced to linear ones, which are significantly more tractable. Second, the transformation provides a deterministic linear relation between the moments of the dynamics. Then, we use tools from functional analysis to derive the result. Considering our earlier findings, we see that this sufficient condition can also be necessary in certain cases, as we demonstrate in Sec. 2.2. Specifically, when unstable oscillations arise in batches with low stability thresholds. Since it takes only one such batch, and the number of possible batches is exponentially large, it is quite likely that SGD operates in this regime.

## 2 WARMUP

In the following, we examine how nonlinear dynamics influence the solutions obtained by gradient-based methods. We begin with GD, showing that the iterates can stably oscillate near an unstable minimum. Combined with step size decay, this suggests that convergence to minima is often mediated by stable oscillations. We then turn to SGD, where we find that, contrary to linear predictions, stability in expectation can be dictated by the worst-case batch rather than an average effect.

### 2.1 NORMAL FORM OF OSCILLATIONS IN GRADIENT DESCENT

A classical result states that GD with constant step size $\eta$ converges to a minimizer in the general case only if it is linearly stable. Specifically, for $\mathcal{L} : \mathbb{R}^d \to \mathbb{R}$ with a minimizer $x^*$, the linear stability threshold is given by $\eta_{\text{lin}} = 2/\lambda_{\max}(\nabla^2 \mathcal{L}(x^*))$, and $x^*$ is linearly stable if and only if $\eta < \eta_{\text{lin}}$. Recently, it has been shown that GD typically operates at the edge of stability (EoS) when optimizing neural networks (Cohen et al., 2021). In this regime, the top eigenvalue of the Hessian hovers just above $2/\eta$ as the parameters approach a minimum. This implies that GD often encounters linearly unstable minima. Although the algorithm cannot converge directly to such points, it can stably oscillate nearby and, after step size decay, eventually converge. Thus, the ability to endure oscillations near a minimum determines whether the algorithm will eventually converge to it.

In this section, we demonstrate that linear stability cannot predict whether GD will stably oscillate near a minimum. To illustrate this, we examine GD's iterates over two univariate functions, depicted in Fig. 1(a), that share the same curvature at the minimum but differ in higher-order terms:

$$f_+(x) = \frac{1}{2}x^2 + \frac{1}{4}x^4, \qquad \text{and} \qquad f_-(x) = \frac{1}{2}x^2 - \frac{1}{4}x^4. \tag{1}$$

Both have the same sharpness at the local minimizer $x^* = 0$, with $f''_+(0) = f''_-(0) = 1$, yielding a linear stability threshold of $\eta_{\text{lin}} = 2$. The iterates of GD are given by

$$x_{t+1} = -(\eta - 1)x_t \pm \eta x_t^3. \tag{2}$$

Let us examine how the asymptotic value of the iterates depends on the step size $\eta$. Figures 1(b) and 1(c) plot the accumulation points of $\{x_t\}$ for various values of $\eta$ on $f_+$ and $f_-$, where $x_0$ is chosen at random from the interval $(-1, 1)$. For $\eta < \eta_{\text{lin}}$, both dynamics converge to $x^* = 0$.

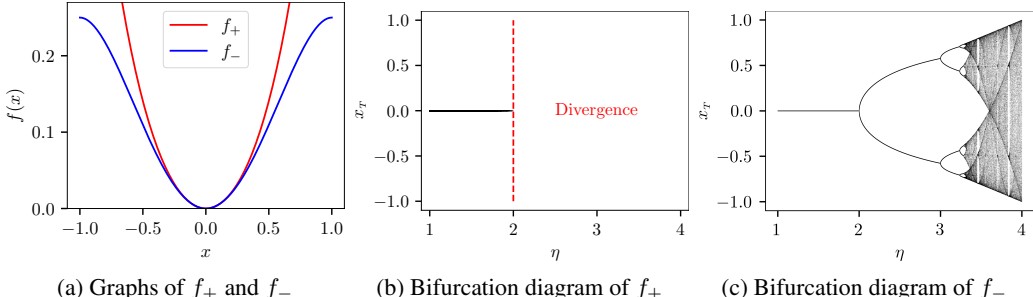

(a) Graphs of $f_+$ and $f_-$  (b) Bifurcation diagram of $f_+$  (c) Bifurcation diagram of $f_-$

Figure 1: **Stable vs. unstable oscillations near a minimum.** We apply GD to $f_+$ and $f_-$ from (1) with various step sizes $\eta \in (1, 4)$. The resulting dynamics (2) correspond to the normal form of a flip bifurcation. Once the step size exceeds the linear stability threshold $\eta_{\text{lin}} = 2$, stability is determined by the sign of the cubic term in the dynamics. Panel (a) shows $f_+$ and $f_-$, whose minima share the same sharpness. Panel (b) visualizes GD's output on $f_+$ with various step sizes. When the step size $\eta$ crosses $\eta_{\text{lin}}$, the minimum $x^* = 0$ loses stability, resulting in unstable oscillations, which lead to divergence. Panel (c) depicts GD's convergent points on $f_-$ for various step sizes. At the threshold, $\eta = \eta_{\text{lin}}$, the minimizer $x^* = 0$ loses stability and the iterates settle into a stable period-2 cycle, which then undergoes period doubling, chaos, and eventually divergence for $\eta > 4$.

However, when $\eta > \eta_{\text{lin}}$, the behavior of the dynamics differs. The iterates on $f_+$ immediately diverge once the step size crosses $\eta_{\text{lin}}$. In contrast, GD on $f_-$ exhibits rich nonlinear dynamics, where it initially settles into stable cycles over a wide range of step sizes while featuring period doubling bifurcations, before transitioning into chaos, and finally diverging once $\eta > 4$. Importantly, when such stable oscillations occur, decaying the step size below $\eta_{\text{lin}}$ results in convergence to $x^*$.

This simple example demonstrates that exceeding the linear stability threshold does not necessarily imply that GD escapes the minimum. Interestingly, under mild assumptions, the behavior of any nonlinear dynamics along the critical manifold near a linearly unstable fixed point can be reduced to this simple one-dimensional map, called *normal form* (see Sec. 5.1). Then, as the example illustrates, the sign of the cubic term in this normal form can be used to determine whether stable oscillations arise. In Sec. 3 we extend this analysis to higher dimensions and derive a general condition for stable oscillations of GD at the edge of stability.

### 2.2 Worst case stability in stochastic gradient descent

Linearized analyses of SGD in expectation predict stability by averaging curvature information across all samples (see App. B). In particular, under mean-square analysis, the distance of the iterates to a minimizer remains bounded as long as the step size $\eta$ is below a threshold determined by an average sharpness of the loss. Here, we show that the full nonlinear dynamics behave differently. Stability in expectation may be governed by the worst-case batch rather than by an average.

To illustrate this discrepancy, we examine the dynamics of SGD on the following functions:

$$f_+(x) = \frac{1}{2}x^2 + \frac{1}{4}x^4, \qquad \text{and} \qquad f_a(x) = \frac{a}{2}x^2, \tag{3}$$

where $a \in (0, 1)$ is a fixed parameter. More specifically, we consider the minimization of the average of $f_+$ and $f_a$, where at each iteration, SGD takes a gradient step with respect to one of these functions, chosen at random. Here $x^* = 0$ is an interpolating minimizer, *i.e.*, it minimizes each function individually. The sharpness of these functions at $x^*$, given by their second derivative, is $h_+ = f_+''(0) = 1$ and $h_a = f_a''(0) = a$. Consequently, the linear stability thresholds for optimizing each function separately are $\eta_+ = 2/h_+ = 2$ and $\eta_a = 2/h_a = 2/a$. Under the linearized mean-square analysis of SGD, the combined stability threshold equals (see App. B)

$$\eta_{\text{lin}} = 2\frac{h_+ + h_a}{h_+^2 + h_a^2} = 2\frac{1 + a}{1 + a^2} > 2. \tag{4}$$

We now compare this prediction with the actual SGD dynamics. Proposition 1 shows that whenever $\eta > 2$, the nonlinear SGD iterates diverge in expectation (see proof in App. C).

**Proposition 1 (Worst case batch)** *Let $\{x_t\}$ be SGD's iterates on $f_+$ and $f_a$ from (3), s.t. $x_0 \neq 0$. If $\eta > 2$ then $\mathbb{E}\big[|x_t - x^*|\big] \underset{t \to \infty}{\longrightarrow} \infty$.*

In other words, because one of the two losses ($f_+$) becomes unstable at $\eta > 2$, the entire stochastic process diverges despite the linearized analysis predicting stability up to $\eta_{\mathrm{lin}} > 2$. This simple example shows that nonlinear SGD can be governed by the least stable batch rather than by an average stability criterion. In Sec.4, we formalize this observation and provide general necessary and sufficient conditions for nonlinear stability of SGD.

## 3 OSCILLATIONS IN GRADIENT DESCENT

In this section, we present a general condition for stable oscillations of GD near a minimum. As noted earlier, GD typically exhibits the edge-of-stability (EoS) phenomenon when optimizing neural networks (Cohen et al., 2021). During the early stages of training, a phase called progressive sharpening, the landscape becomes sharper as the top eigenvalue of the Hessian increases until it reaches the linear stability threshold of $2/\eta$ (Wang et al., 2022). Beyond this point, the sharpness remains slightly above $2/\eta$ for the rest of the training. Consequently, as the iterates approach a minimum, GD often encounters minima whose sharpness marginally exceeds the linear stability threshold. While direct convergence to such minimizers is impossible, the iterates can stably oscillate in their vicinity. Then, once the step size decays, these oscillations vanish, allowing the method to settle into the minimum. Thus, understanding the behavior of GD at the edge of stability in the vicinity of minima is critical for determining to which minima it converges. Here we have the following result, which uses the $k$th order derivative in multilinear form, denoted as $\mathcal{D}^k$.

**Theorem 1 (Stable oscillations)** *Let $\mathcal{L} : \mathbb{R}^d \to \mathbb{R}$ and $\boldsymbol{x}^*$ be its local minimizer, such that $\mathcal{L}$ is four times differentiable at $\boldsymbol{x}^*$. Assume $\nabla^2\mathcal{L}(\boldsymbol{x}^*)$ is strictly positive and let $\boldsymbol{v}$ be a top eigenvector corresponding to the maximal eigenvalue. Suppose GD on $\mathcal{L}$ with step size $\eta$ operates at the edge of stability, i.e., $\lambda_{\max}\big(\nabla^2\mathcal{L}(\boldsymbol{x}^*)\big) = 2/\eta$, and that $\lambda_{\max}$ has multiplicity one. Then a stable period-2 cycle exists at the vicinity of $\boldsymbol{x}^*$ if and only if*

$$\mathcal{D}^3\mathcal{L}(\boldsymbol{x}^*)[\boldsymbol{v}, \boldsymbol{v}, \boldsymbol{q}] > \mathcal{D}^4\mathcal{L}(\boldsymbol{x}^*)[\boldsymbol{v}, \boldsymbol{v}, \boldsymbol{v}, \boldsymbol{v}], \tag{5}$$

*where*

$$\boldsymbol{q} \triangleq \big[\nabla^2\mathcal{L}(\boldsymbol{x}^*)\big]^{-1}\nabla_{\boldsymbol{v}}\mathcal{D}^3\mathcal{L}(\boldsymbol{x}^*)[\boldsymbol{v}, \boldsymbol{v}, \boldsymbol{v}]. \tag{6}$$

This theorem states that GD can stably oscillate near a minimum if and only if the condition in (5) holds. This condition is composed of high-order derivatives of the loss. Intuitively, it suggests that when the third derivative dominates over the fourth across the sharpest direction, we have stable oscillations and vice versa. The expression for $\boldsymbol{q}$ has a Newton-like structure, where the inverse Hessian is applied to a gradient. However, this gradient acts only on the cubic term in the Taylor expansion of the loss, not on the full objective. Obviously, $\nabla_{\boldsymbol{v}}\mathcal{D}^3\mathcal{L}(\boldsymbol{x}^*)[\boldsymbol{v}, \boldsymbol{v}, \boldsymbol{v}]$ equals $3\mathcal{D}^3\mathcal{L}(\boldsymbol{x}^*)[\boldsymbol{v}, \boldsymbol{v}]$, and thus $\boldsymbol{v}$'s scale and polarity do not affect the condition. When the condition is satisfied, step sizes slightly above $2/\lambda_{\max}$ produce stable periodic oscillations, whose amplitude grows with $\eta$, while smaller step sizes converge to the minimum. Conversely, if the condition is not met, any step size larger than $2/\lambda_{\max}$ leads the iterates to escape the small neighborhood of the minimum. The proof outline, along with additional information about bifurcations, are given in Sec. 5.

To illustrate Thm. 1, we consider a simple example, shown in Fig. 2. Let

$$f_\alpha(x) = \frac{1}{2}x^2 + \frac{\alpha}{6}x^3 + \frac{1}{8}x^4, \tag{7}$$

with minimizer at $x^* = 0$. Figure 2(a) depicts $f_\alpha$ near the minimum for a few values of $\alpha$. The linear stability threshold of GD around $x^*$ is $\eta_{\mathrm{lin}} = 2$, at which the update rule becomes

$$x_{t+1} = -x_t - \alpha x_t^2 - x_t^3. \tag{8}$$

In this case, the condition for stable oscillations (5) simplifies to $|\alpha| > 1$ (see App. E). Figure 2(b) presents the accumulation points of the iterates for a range of $\alpha$. When $|\alpha| > 1$, GD's iterates converge to stable period-2 cycles, whereas for $|\alpha| < 1$, the iterates diverge. This example demonstrates that Thm. 1 captures the precise phase transition from stable to unstable oscillations.

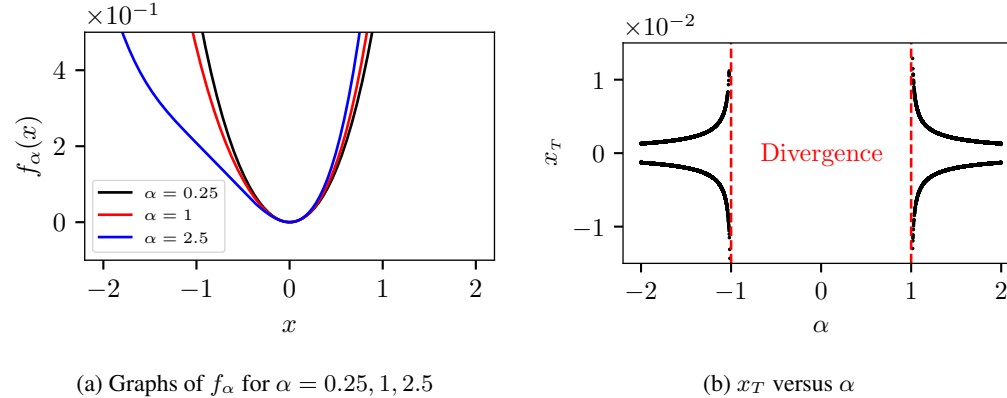

(a) Graphs of $f_\alpha$ for $\alpha = 0.25, 1, 2.5$        (b) $x_T$ versus $\alpha$

**Figure 2: Demonstration of Thm. 1.** Consider $f_\alpha(x) = \frac{1}{2}x^2 + \frac{\alpha}{6}x^3 + \frac{1}{8}x^4$, whose linear stability threshold under GD is $\eta_{\text{lin}} = 2$. At this step size, the update rule becomes $x_{t+1} = -x_t - \alpha x_t^2 - x_t^3$. According to Thm. 1, GD oscillates stably around the minimum $x^* = 0$ if and only if $|\alpha| > 1$ (see App. E). Panel (a) plots $f_\alpha$ near $x^*$ for three choices of $\alpha$, highlighting the asymmetry introduced by the cubic term. Panel (b) shows the long-term value $x_T$ across a range of $\alpha$. When $|\alpha| > 1$, GD converges to a stable period-2 cycle, whereas for $|\alpha| < 1$ the iterates diverge. This confirms that condition (5) precisely captures the transition from stability to instability.

## 4 STABILITY OF NONLINEAR DYNAMICS IN SGD

In this section, we present our results on the stability of nonlinear dynamics in SGD. Let $f_i : \mathbb{R}^d \to \mathbb{R}$ be analytic for all $i \in [n]$. We define the loss function and its batch approximation as

$$\mathcal{L}(\boldsymbol{x}) = \frac{1}{n}\sum_{i=1}^n f_i(\boldsymbol{x}), \qquad \text{and} \qquad \hat{\mathcal{L}}_\mathcal{B}(\boldsymbol{x}) = \frac{1}{B}\sum_{i\in\mathcal{B}} f_i(\boldsymbol{x}), \qquad (9)$$

where $\mathcal{B} \subseteq [n]$ is a batch (set) of size $|\mathcal{B}| = B$. The iterates of SGD are given by

$$\boldsymbol{x}_{t+1} = \boldsymbol{x}_t - \eta\nabla\hat{\mathcal{L}}_{\mathcal{B}_t}(\boldsymbol{x}_t). \qquad (10)$$

Here, $\mathcal{B}_t$ refers to a stochastic batch sampled at iteration $t$. We assume that the batches $\{\mathcal{B}_t\}$ are drawn without replacement, independently across iterations. Namely, there are distinct samples within each batch and possible repetitions between different batches.

Our analysis focuses on the dynamics of SGD near interpolating minimizers. This setting has been extensively studied by prior work, particularly in the context of dynamical stability and overparameterized models (Wu et al., 2018; Ma & Ying, 2021; Mulayoff & Michaeli, 2024).

**Definition 1 (Interpolating minimizer)** *We say that $\boldsymbol{x}^* \in \mathbb{R}^d$ is an interpolating minimizer of $\mathcal{L}$ if*

$$\forall i \in [n] \qquad \nabla f_i(\boldsymbol{x}^*) = \boldsymbol{0} \qquad \text{and} \qquad \nabla^2 f_i(\boldsymbol{x}^*) \succ \boldsymbol{0} \text{ (positive definite)}. \qquad (11)$$

To gain intuition about the stability of SGD near interpolating minimizers, it is useful to examine the dynamics of the iterates across all possible batches. Concretely, consider running GD separately on every batch. For a given step size, some batches may converge, while others may exhibit stable oscillations or even diverge. To capture the average behavior of the algorithm, we adopt the notion of stability in expectation (Ma & Ying, 2021). A popular instance of this approach is the mean-square (Wu et al., 2018), whose stability threshold in the linear setting aggregates curvature information from all samples (Mulayoff & Michaeli, 2024). However, as shown in Sec. 2, nonlinear dynamics behave differently. Instability of even a single batch can be enough to cause the mean to diverge.

**Theorem 2 (Necessary condition)** *Let $\boldsymbol{x}^*$ be an interpolating minimizer of $\mathcal{L}$, $\boldsymbol{x}_0 \in \mathbb{R}^d$, and $\mathcal{B}_*$ be a batch of size $B$. Denote GD's iterates with step size $\eta$ over $\hat{\mathcal{L}}_{\mathcal{B}_*}$ by $\boldsymbol{x}_t^{(\mathcal{B}_*)}$. If*

$$\sqrt[t]{\left\|\boldsymbol{x}_t^{(\mathcal{B}_*)} - \boldsymbol{x}^*\right\|} \xrightarrow[t\to\infty]{} \infty, \qquad (12)$$

*then SGD's iterates $\{\boldsymbol{x}_t\}$ of (10) with step size $\eta$ diverge in expectation, i.e., $\mathbb{E}\big[\|\boldsymbol{x}_t - \boldsymbol{x}^*\|\big] \xrightarrow[t\to\infty]{} \infty$.*

In simple terms, this theorem states that if GD on even a single batch diverges at a rate higher than linear, then SGD as a whole will also diverge in expectation (see proof in App. F). Section 2.1 provides a concrete example, where GD's iterates on the function $f_+$ in (1) diverge superlinearly (see App. C). Consequently, if the finite-sum loss $\mathcal{L}$ contains a batch loss $\hat{\mathcal{L}}_B = f_+$, then SGD will diverge in expectation. This is the underlying principle behind the observation in Sec. 2.2.

What can we learn from this result? Suppose the iterates reach a neighborhood of a minimizer $\boldsymbol{x}^*$, and let $\eta_{\mathcal{B}} = 2/\lambda_{\max}(\nabla^2 \hat{\mathcal{L}}_{\mathcal{B}}(\boldsymbol{x}^*))$ denote the linear stability threshold of a batch loss $\hat{\mathcal{L}}_B$. Clearly, for small enough neighbourhood, GD can diverge only if the step size satisfies $\eta \geq \eta_{\mathcal{B}}$. Notably, if the condition for stable oscillations in Thm. 1 is violated, then superlinear divergence may already occur at the threshold $\eta = \eta_{\mathcal{B}}$. In this case, the stability threshold of SGD is effectively capped by $\eta_{\mathcal{B}}$. This naturally raises the following question. Under what conditions can we guarantee the stability of SGD? The result below addresses this point (see proof in App. 5.2).

**Theorem 3 (Sufficient condition)** *Let $\boldsymbol{x}^*$ be an interpolating minimizer of $\mathcal{L}$, and consider SGD's iterates* (10) *denoted by $\{\boldsymbol{x}_t\}$. If*

$$\eta < \min_{\mathcal{B}:|\mathcal{B}|=B} \frac{2}{\lambda_{\max}\big(\nabla^2 \hat{\mathcal{L}}_{\mathcal{B}}(\boldsymbol{x}^*)\big)}, \tag{13}$$

*then there exists a neighborhood $\{\boldsymbol{x}_0 : \|\boldsymbol{x}_0 - \boldsymbol{x}^*\| < \rho\}$ s.t. $\mathbb{E}\big[\|\boldsymbol{x}_t - \boldsymbol{x}^*\|_k^k\big]\rho^{-k} \underset{t\to\infty}{\longrightarrow} 0$ for all even $k$.*

This result shows that if the step size is linearly stable with respect to all batches, then the full nonlinear dynamics of SGD are stable in expectation. As demonstrated in Sec. 2.2, this sufficient condition can also be necessary in certain cases. Specifically, when unstable oscillations leading to superlinear divergence arise in batches with low linear stability thresholds. Since it takes only one such batch, and the number of possible batches is exponentially large, it is quite likely that SGD operates in this regime.

# 5 DERIVATIONS

## 5.1 GRADIENT DESCENT OSCILLATIONS AS A FLIP BIFURCATION

In this section, we give a brief review of bifurcations and formulate GD's dynamics in this framework. For a comprehensive overview of bifurcations, see Kuznetsov (1998). Consider the parameter-dependent nonlinear system $\boldsymbol{x}_{t+1} = \boldsymbol{\psi}(\boldsymbol{x}_t, \eta)$ with fixed point $\boldsymbol{x}^*$, *i.e.*, $\boldsymbol{\psi}(\boldsymbol{x}^*, \eta) = \boldsymbol{x}^*$. In general, bifurcations of fixed points occur when a parameter changes its value, while affecting the stability of the dynamics. The special case of flip bifurcation, also called period-doubling, happens when the fixed point $\boldsymbol{x}^*$ loses stability as the parameter $\eta$ changes, and a period-2 cycle emerges. Mathematically, let us define the critical value of the parameter $\eta_c$ such that the dominant[1] eigenvalue of the Jacobian $\mathcal{D}_{\boldsymbol{x}}\boldsymbol{\psi}(\boldsymbol{x}^*, \eta_c)$ equals $-1$. Then a flip bifurcation takes place while this eigenvalue crosses minus one on the real line as $\eta$ exceeds $\eta_c$. Here $\eta_c$ is the linear stability threshold.

In this case, for $\eta < \eta_c$, the fixed point $\boldsymbol{x}^*$ is stable, and if the iterates happen to arrive close by, they will be attracted to it. If the Jacobian has full rank, the iterates will in fact converge to $\boldsymbol{x}^*$. However, when $\eta$ is slightly above $\eta_c$, the fixed point $\boldsymbol{x}^*$ is no longer stable, and a period-2 cycle appears as

$$\boldsymbol{\psi}\big(\boldsymbol{x}^{(1)}, \eta\big) = \boldsymbol{x}^{(2)}, \qquad \text{and} \qquad \boldsymbol{\psi}\big(\boldsymbol{x}^{(2)}, \eta\big) = \boldsymbol{x}^{(1)}. \tag{14}$$

The stability of the resulting period-2 cycle is governed by the coefficient of the cubic term in the corresponding normal form of the bifurcation. This form provides a canonical (standard) dynamics to which any flip bifurcation can be reduced. Concretely, consider the dynamics along the one-dimensional critical manifold, tangent to the dominant eigenvector of the Jacobian. Then this dynamics can be transformed into (Kuznetsov, 1998, Sec 5.4)

$$\xi_{t+1} = -\xi_t + C_0\xi_t^3 + O\big(\xi_t^4\big), \tag{15}$$

where $C_0$ is the first Lyapunov coefficient. When $C_0 > 0$, the resulting cycle is stable (supercritical bifurcation), and the dynamics in the long run will alternate between $\boldsymbol{x}^{(1)}$ and $\boldsymbol{x}^{(2)}$. Whereas for

---

[1]Dominant eigenvalue is an eigenvalue that has maximal absolute value. Here we assume that it is unique.

$C_0 < 0$, the cycle is unstable (subcritical bifurcation) and the iterates will diverge from $\boldsymbol{x}^*$. The expression for $C_0$, involving the second- and third-order derivatives of $\boldsymbol{\psi}$ at $\boldsymbol{x}^*$, is given in App. D.

We now turn to apply this theory to prove Thm. 1. In the context of GD's iterates near a minimum, the dynamics evolve according to the update rule

$$\boldsymbol{x}_{t+1} = \boldsymbol{x}_t - \eta \nabla \mathcal{L}(\boldsymbol{x}_t) \triangleq \boldsymbol{\psi}_t(\boldsymbol{x}_t, \eta), \tag{16}$$

where $\mathcal{L}$ is an objective function to be minimized, and $\eta$ is the step size. Obviously, minimizers of $\mathcal{L}$ are the fixed points of $\boldsymbol{\psi}$, as the gradient vanishes at these points. The Jacobian of the GD map is

$$\mathcal{D}_x \boldsymbol{\psi}(\boldsymbol{x}, \eta) = \boldsymbol{I} - \eta \nabla^2 \mathcal{L}(\boldsymbol{x}). \tag{17}$$

Note that the eigenvalues of the Jacobian are given by $\{1 - \eta \lambda_i(\nabla^2 \mathcal{L})\}$. Let $\boldsymbol{x}^*$ be a minimizer of $\mathcal{L}$, then the critical value of $\eta$ is the well known linear stability threshold

$$\eta_{\lin} = \frac{2}{\lambda_{\max}\left(\nabla^2 \mathcal{L}(\boldsymbol{x}^*)\right)}. \tag{18}$$

Thus, as $\eta$ exceeds $\eta_{\lin}$, the dominant eigenvalue of the Jacobian crosses $-1$ on the real axis, matching the scenario of the flip bifurcation. Assuming $\nabla^2 \mathcal{L}(\boldsymbol{x}^*)$ is strictly positive and $\lambda_{\max}(\nabla^2 \mathcal{L}(\boldsymbol{x}^*))$ has multiplicity one, the stability of oscillations in the small neighborhood of $\boldsymbol{x}^*$ is governed by $C_0$. Overall, we see that oscillations near a minimum $\boldsymbol{x}^*$ are stable if and only if $C_0$ is positive. In App. D we show that a positive Lyapunov coefficient is equivalent to the condition in (5).

## 5.2 SUFFICIENT CONDITION FOR STABILITY OF SGD

In this section, we derive Thm. 3. SGD update rule with step size $\eta$ is given by

$$\boldsymbol{x}_{t+1} = \boldsymbol{x}_t - \eta \nabla \hat{\mathcal{L}}_{\mathcal{B}_t}(\boldsymbol{x}_t) \triangleq \hat{\boldsymbol{\psi}}_{\mathcal{B}_t}(\boldsymbol{x}_t), \tag{19}$$

where $\hat{\boldsymbol{\psi}}_{\mathcal{B}_t} : \mathbb{R}^d \to \mathbb{R}^d$ is the SGD map. As $\boldsymbol{x}^*$ is an interpolating minimizer of $\mathcal{L}$, we have

$$\hat{\boldsymbol{\psi}}_{\mathcal{B}_t}(\boldsymbol{x}^*) = \boldsymbol{x}^* \text{ w.p. } 1. \tag{20}$$

Since $\{f_i\}$ are analytic, we can use the Taylor expansions of $\hat{\boldsymbol{\psi}}_{\mathcal{B}_t}$ at $\boldsymbol{x}^*$ to get

$$\hat{\boldsymbol{\psi}}_{\mathcal{B}_t}(\boldsymbol{x}) = \hat{\boldsymbol{\psi}}_{\mathcal{B}_t}(\boldsymbol{x}^*) + \sum_{k=1}^{\infty} \frac{1}{k!} \mathcal{D}^k \hat{\boldsymbol{\psi}}_{\mathcal{B}_t}(\boldsymbol{x}^*)(\boldsymbol{x} - \boldsymbol{x}^*)^{\otimes k}, \tag{21}$$

where $\mathcal{D}^k$ is the $k$th order derivative in *matrix form* (not to be confused with $\mathcal{D}^k$). Let

$$\Delta \boldsymbol{x}_t^k \triangleq (\boldsymbol{x}_t - \boldsymbol{x}^*)^{\otimes k} \in \mathbb{R}^{d^k} \tag{22}$$

be the $k$th Kronecker power of the distance to the minimum[2]. Then from the update rule in (19)

$$\mathbb{E}\left[\Delta \boldsymbol{x}_{t+1}\right] = \mathbb{E}\left[\sum_{k=1}^{\infty} \frac{1}{k!} \mathcal{D}^k \hat{\boldsymbol{\psi}}_{\mathcal{B}_t}(\boldsymbol{x}^*) \Delta \boldsymbol{x}_t^k\right] = \sum_{k=1}^{\infty} \frac{1}{k!} \mathbb{E}\left[\mathcal{D}^k \hat{\boldsymbol{\psi}}_{\mathcal{B}_t}(\boldsymbol{x}^*)\right] \mathbb{E}\left[\Delta \boldsymbol{x}_t^k\right], \tag{23}$$

where we used the fact that $\{\mathcal{D}^k \hat{\boldsymbol{\psi}}_{\mathcal{B}_t}(\boldsymbol{x}^*)\}_{k=1}^{\infty}$ and $\boldsymbol{x}_t$ are statistically independent. We see that the evolution of the first moment of the distance to the minimum, $\mathbb{E}[\Delta \boldsymbol{x}]$, depends *linearly* on all higher-order moments $\{\mathbb{E}[\Delta \boldsymbol{x}^k]\}_{k=1}^{\infty}$. Consequently, analyzing the stability of SGD in expectation requires studying the joint dynamics of all moments. In App. H, we show that the evolution of the $k$th moment over time is

$$\mathbb{E}\left[\Delta \boldsymbol{x}_{t+1}^k\right] = \mathbb{E}\left[(\Delta \boldsymbol{x}_{t+1})^{\otimes k}\right] = \mathbb{E}\left[\left(\sum_{p=1}^{\infty} \frac{1}{p!} \mathcal{D}^p \hat{\boldsymbol{\psi}}_{\mathcal{B}_t}(\boldsymbol{x}^*) \Delta \boldsymbol{x}_t^p\right)^{\otimes k}\right] = \sum_{p=k}^{\infty} \boldsymbol{\Psi}_{k,p} \mathbb{E}\left[\Delta \boldsymbol{x}_t^p\right], \tag{24}$$

where explicit expression for $\boldsymbol{\Psi}_{k,p} \in \mathbb{R}^{d^k \times d^p}$ is given in App. H. Once again, we obtain a linear relation between the moments at successive times. This motivates us to express the mapping from $\{\mathbb{E}[\Delta \boldsymbol{x}_t^k]\}_{k=1}^{\infty}$ to $\{\mathbb{E}[\Delta \boldsymbol{x}_{t+1}^k]\}_{k=1}^{\infty}$ as a linear operator on the infinite-dimensional Hilbert space $\ell_2$.

---

[2]The first power $\Delta \boldsymbol{x}_t^1$ is denoted simply $\Delta \boldsymbol{x}_t$.

This formulation is meaningful only if the sequence has finite norm and the operator is bounded. To ensure this, we introduce a radius $\rho > 0$ and analyze a scaled version of the moments. Let

$$\bar{\boldsymbol{\mu}}_t^k \triangleq \mathbb{E}\left[\left(\frac{\boldsymbol{x}_t - \boldsymbol{x}^*}{\rho}\right)^{\otimes k}\right] = \rho^{-k}\mathbb{E}\left[\Delta\boldsymbol{x}_t^k\right]. \tag{25}$$

Therefore,

$$\bar{\boldsymbol{\mu}}_{t+1}^k = \rho^{-k}\mathbb{E}\left[\Delta\boldsymbol{x}_{t+1}^k\right] = \sum_{p=k}^{\infty}\rho^{-k}\boldsymbol{\Psi}_{k,p}\mathbb{E}\left[\Delta\boldsymbol{x}_t^p\right] = \sum_{p=k}^{\infty}\rho^{p-k}\boldsymbol{\Psi}_{k,p}\mathbb{E}\left[\rho^{-p}\Delta\boldsymbol{x}_t^p\right] = \sum_{p=k}^{\infty}\rho^{p-k}\boldsymbol{\Psi}_{k,p}\bar{\boldsymbol{\mu}}_t^k. \tag{26}$$

Define the linear operator $\boldsymbol{\Psi}_\rho$ in Hilbert space $\ell_2$ and the moments vector $\bar{\boldsymbol{\mu}}_t$ as

$$\bar{\boldsymbol{\mu}}_t = \begin{bmatrix} \bar{\boldsymbol{\mu}}_t^1 \\ \bar{\boldsymbol{\mu}}_t^2 \\ \bar{\boldsymbol{\mu}}_t^3 \\ \vdots \end{bmatrix} \quad \text{and} \quad \boldsymbol{\Psi}_\rho = \begin{bmatrix} \boldsymbol{\Psi}_{1,1} & \rho\boldsymbol{\Psi}_{1,2} & \rho^2\boldsymbol{\Psi}_{1,3} & \cdots \\ \boldsymbol{0} & \boldsymbol{\Psi}_{2,2} & \rho\boldsymbol{\Psi}_{2,3} & \cdots \\ \boldsymbol{0} & \boldsymbol{0} & \boldsymbol{\Psi}_{3,3} & \cdots \\ \vdots & \vdots & \vdots & \ddots \end{bmatrix}, \tag{27}$$

then

$$\bar{\boldsymbol{\mu}}_{t+1} = \boldsymbol{\Psi}_\rho\bar{\boldsymbol{\mu}}_t. \tag{28}$$

This relation is valid only when $\boldsymbol{\Psi}_\rho$ is bounded. Intuitively, taking smaller values of $\rho$ can help bound the operator. Assuming the operator is bounded, (28) unfolds as $\bar{\boldsymbol{\mu}}_t = \boldsymbol{\Psi}_\rho^t\bar{\boldsymbol{\mu}}_0$. To impose a condition on the initial point $\boldsymbol{x}_0$, observe that $\bar{\boldsymbol{\mu}}_0 \in \ell_2$, and thus must be square-summable. Hence,

$$\|\bar{\boldsymbol{\mu}}_0\|^2 = \sum_{k=1}^{\infty}\|\bar{\boldsymbol{\mu}}_0^k\|^2 = \sum_{k=1}^{\infty}\left\|\left(\frac{\boldsymbol{x}_0 - \boldsymbol{x}^*}{\rho}\right)^{\otimes k}\right\|^2 = \sum_{k=1}^{\infty}\left(\frac{\|\boldsymbol{x}_0 - \boldsymbol{x}^*\|}{\rho}\right)^{2k}. \tag{29}$$

The above expression is finite if and only if $\|\boldsymbol{x}_0 - \boldsymbol{x}^*\| < \rho$, which defines the neighborhood around the minimum where our analysis applies. We see that choosing a smaller $\rho$ to ensure boundedness of the operator correspondingly shrinks this neighborhood.

For stable dynamics in expectation, the linear system in (28) must be stable. Note that under $\boldsymbol{\Psi}_\rho$, moment vectors map naturally to moment vectors. Thus, to get the exact stability threshold, we would need the response of $\boldsymbol{\Psi}_\rho$ to be smaller than one on this restricted set. Instead, we relax this constraint to obtain a sufficient condition, requiring stability for any vector in $\ell_2$. In App. I, we prove that under the condition (13), there exists a value of $\rho > 0$ ensuring boundedness of the operator. Then, in App. K we show that once the operator is bounded, its spectral radius is strictly less than one. Therefore, $\|\bar{\boldsymbol{\mu}}_t\| \to 0$ as $t$ tends to infinity (see App. G). Hence, each normalized moment also tends to zero elementwise, *i.e.*, $\bar{\boldsymbol{\mu}}_t^k \to \boldsymbol{0}$. Since $\bar{\boldsymbol{\mu}}_t^k$ contains all degree-$k$ monomials of the components of $\Delta\boldsymbol{x}_t$, denoted $\{\Delta\boldsymbol{x}_{t,i}\}_{i=1}^d$, summing over the subset of single-variable terms we get

$$\sum_{i=1}^{d}\mathbb{E}\left[(\Delta\boldsymbol{x}_{t,i})^k\right] = \mathbb{E}\left[\sum_{i=1}^{d}(\boldsymbol{x}_{t,i} - \boldsymbol{x}_i^*)^k\right]\rho^{-k} \xrightarrow[t\to\infty]{} 0. \tag{30}$$

Restricting this to even order moments (even $k$), we get $\mathbb{E}\left[\|\boldsymbol{x}_t - \boldsymbol{x}^*\|_k^k\right]\rho^{-k} \xrightarrow[t\to\infty]{} 0$.

# 6 RELATED WORK

**Bifurcation, oscillations and EoS in GD.** Cohen et al. (2021) examined the behavior of GD, and showed that it typically happens at the edge of stability. Wang et al. (2022) proved progressive sharpening for a two-layer network and analyzed the EOS dynamics through four phases, depending on the change in the sharpness value. Zhu et al. (2022) gave a simple example that exhibits EoS. Ma et al. (2022) analyzed EoS under the assumption of subquadratic growth of the loss. Ahn et al. (2022) illustrated that unstable convergence is possible in specific cases. Damian et al. (2023) showed how GD self-stabilizes. Specifically, they demonstrated that during the momentary divergence of the iterates along the sharpest eigenvector direction of the Hessian, the iterates also move along the negative direction of the gradient of the curvature, which leads to stabilizing the sharpness to $2/\eta$. Kreisler et al. (2023); Song & Yun (2023) proved that under EoS, different GD trajectories align on a specific bifurcation diagram independent of initialization. Ghosh et al. (2025) analyzed the dynamics of deep linear networks, focusing on 2-period cycle, while showing that oscillations occur within a small subspace, where the dimension of the subspace is controlled by the step size. Chen et al. (2024) studied GD dynamics on quadratic loss from stability up to the chaos phase.

**Stability of SGD.** Empirically, Keskar et al. (2016); Jastrzębski et al. (2017); Jastrzębski et al. (2019; 2020) have shown that SGD with a large step size or small batch size leads to flatter minima. Cohen et al. (2021, App. G) found that with large batches, the sharpness behaves similarly to full-batch gradient descent. Gilmer et al. (2022) studied how the curvature of the loss affects the training dynamics in multiple settings.

On the theoretical side, Wu et al. (2018) analyzed stability in the mean-square sense and provided an implicit sufficient condition. Granziol et al. (2022) used random matrix theory to characterize the maximal stable learning rate as a function of batch size, under certain assumptions on Hessian noise. Velikanov et al. (2023) studied SGD with momentum and derived an implicit upper bound on the learning rate using spectrally expressible approximations and a moment-generating function. Ma & Ying (2021) investigated higher-order moments of SGD and established an implicit necessary and sufficient stability condition. Wu et al. (2022) proposed a necessary condition based on an alignment property, though a general analytic bound for this property is missing. Ziyin et al. (2023) examined stability in probability rather than in mean square, showing that SGD can in theory converge with high probability to linearly unstable minima for GD, *i.e.*, where $\eta \gg 2/\lambda_{\max}(\nabla^2 \mathcal{L})$. However, this prediction was not observed empirically. Mulayoff et al. (2021) considered non-differentiable minima and derived a necessary condition for strong stability, meaning SGD remains within a ball around the minimum. Finally, Mulayoff & Michaeli (2024) provided the exact stability criterion explicitly in closed-form expression for the linearized dynamics.

Additionally, Liu et al. (2021) analyzed the covariance matrix of the stationary distribution of iterates near minima, and Ziyin et al. (2022) extended these results by deriving an implicit relation between this covariance and that of the gradient noise. However, both works leave open the question of when the dynamics actually converge to a stationary state. Recently, Lee & Jang (2023) examined the stability of SGD along its trajectory and established an explicit exact condition for objective decrease via a descent lemma in expectation.

# 7 CONCLUSION, LIMITATIONS AND FUTURE DIRECTIONS

In this paper, we investigated the nonlinear stability of gradient descent GD and SGD. For GD, we derived an explicit condition characterizing when stable oscillations arise at the edge of stability, namely, when the cubic term dominates the quartic term in the local Taylor expansion of the objective. For SGD, we showed that the instability of even a single batch can be sufficient to render the entire dynamics unstable in expectation, implying that stability is dictated by the worst-case batch rather than by an average effect. Finally, we proved that if the step size is stable with respect to all batches, then all moments of the full nonlinear SGD dynamics remain stable in a neighborhood of the minimizer. Together, these results reveal that nonlinear effects can fundamentally reshape the stability landscape compared with standard linear analyses.

**Limitations and future directions.** Our analysis of oscillations in GD focuses on isolated minima, where the Jacobian of the dynamical system has a single critical eigenvalue. In deep learning, however, minima often form low-dimensional manifolds with multiple near-critical directions. This can lead to richer local dynamics, including combinations of fold and flip behaviors, and a more complex stability picture. Extending our results to this setting, potentially via generalized fold-flip bifurcations (Kuznetsov et al., 2004), is an important direction for future work.

For our analysis of SGD we assume an interpolating setting in which all mini-batches share the same minimizer. This allows us to prove that stability of each batch implies stability of the full dynamics. In practice, however, batches may have distinct or only approximately aligned minima. In such cases, SGD cannot converge exactly to the minimizer; even if the dynamics are stable, the algorithm exhibits an inherent bias in expectation (Défossez & Bach, 2015). In this work we adopt a convergence-in-expectation perspective for SGD, but in non-interpolating settings the limiting point may be biased or correspond to a different minimum. Developing a principled notion of nonlinear stability that captures this behavior remains an important direction for future research.

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

# APPENDICES

## A LARGE LANGUAGE MODEL (LLM) USAGE DISCLOSURE

In this paper, we used LLMs only to polish the text. The LLM was given a text written by the authors, and it suggested alternative ways of writing. These suggestions, if accepted, were further refined by the authors and not used as is. Specifically, LLMs were *not* used to generate any text from scratch, or to suggest research direction, or to derive the results.

## B BACKGROUND ON LINEAR STABILITY OF SGD

Analyzing the full dynamics of SGD can be hard. Therefore, many works opt to study the linearized dynamic near minima (Wu et al., 2018; Ma & Ying, 2021; Mulayoff et al., 2021; Mulayoff & Michaeli, 2024), as it is common in the analysis of nonlinear systems. In our paper we focus on interpolating minimizers, defined in Def. 1. In this case, the linearized dynamics is defined below.

**Definition 2 (Linearized dynamics)** *Let $\mathcal{L}$ form (9), and $\boldsymbol{x}^*$ be its interpolating minimizer, s.t. $\mathcal{L}$ is twice differentiable at $\boldsymbol{x}^*$. Then the linearized dynamics of SGD near $\boldsymbol{x}^*$ are given by*

$$\tilde{\boldsymbol{x}}_{t+1} = \tilde{\boldsymbol{x}}_t - \frac{\eta}{B} \sum_{i \in \mathcal{B}_t} \nabla^2 f_i(\boldsymbol{x}^*)(\tilde{\boldsymbol{x}}_t - \boldsymbol{x}^*). \tag{31}$$

The linearized dynamics can be viewed as SGD on the second-order approximation of $\mathcal{L}$ at $\boldsymbol{x}^*$,

$$\tilde{\mathcal{L}}(\boldsymbol{x}) = \mathcal{L}(\boldsymbol{x}^*) + \frac{1}{2}(\boldsymbol{x} - \boldsymbol{x}^*)^{\mathrm{T}} \nabla^2 \mathcal{L}(\boldsymbol{x}^*)(\boldsymbol{x} - \boldsymbol{x}^*). \tag{32}$$

Therefore, linear dynamics analysis is exact only when $\{f_i\}$ are all quadratic potentials.

There are few traditional ways to define the convergence of random processes, such as the iterates of SGD. One prominent choice is to use the mean square sense of convergence to define stability. For univariate optimization, the mean square linear stability threshold is as follows. Generalization to higher dimensions and non-interpolating minima can be found in Mulayoff & Michaeli (2024).

**Theorem 4 (Univariate linear stability threshold, Wu et al. (2018))** *Let $f_i : \mathbb{R} \to \mathbb{R}$ be twice differentiable functions and let $x^*$ be an interpolating minimum of the loss, i.e.,*

$$\forall\, 1 \le i \le n \qquad f_i'(x^*) = 0 \qquad and \qquad h_i \triangleq f_i''(x^*) > 0. \tag{33}$$

*Define*

$$h = \frac{1}{n} \sum_{i=1}^{n} h_i, \qquad s^2 = \frac{1}{n} \sum_{i=1}^{n} (h_i - h)^2, \qquad and \qquad p = \frac{n - B}{B(n-1)}. \tag{34}$$

*Consider the iterates of the linearized SGD $\{\tilde{x}_t\}$ in (31). Then, $\mathbb{E}[(\tilde{x}_t - x^*)^2]$ is bounded if and only if $\eta \le \eta_{\mathrm{lin}}$, where*

$$\eta_{\mathrm{lin}} \triangleq \frac{2h}{h^2 + ps^2}. \tag{35}$$

From this result, we see that the linear stability threshold $\eta_{\mathrm{lin}}$ takes into account the sharpness of all functions $\{f_i\}$. When the batch size $B$ equals one, we get $p = 1$ and then

$$\eta_{\mathrm{lin}} = \frac{2h}{h^2 + s^2} = 2 \frac{\sum_{i=1}^{n} h_i}{\sum_{i=1}^{n} h_i^2}. \tag{36}$$

## C PROOF OF PROPOSITION 1

Let

$$f_+(x) = \frac{1}{2} x^2 + \frac{1}{4} x^4. \tag{37}$$

Here we show that when GD is applied on $f_+$ with step size $\eta > 2$, its iterates diverge in rate higher than linear. In this case, Thm. 2 tell us that SGD dynamics also diverge.

The GD map on $f_+$ is

$$\psi(x) = x - \eta f'_+(x) = (1 - \eta)x - \eta x^3. \tag{38}$$

Define

$$\psi^t = \underbrace{\psi \circ \cdots \circ \psi \circ \psi}_{t \text{ times}}. \tag{39}$$

Assume that $\eta > 2$, and note that for all $x \in \mathbb{R}$

$$|\psi(x)| = |(1 - \eta)x - \eta x^3| = (\eta - 1)|x| + \eta |x|^3 = |\psi(|x|)|. \tag{40}$$

Let $\tilde{\psi}(x) = |\psi|(|x|)$, then

$$|\psi^t(x_0)| = \tilde{\psi}^t(|x_0|). \tag{41}$$

Since $\tilde{\psi} : \mathbb{R}_+ \to \mathbb{R}_+$ is monotonically increasing on $\mathbb{R}_+$, we have that its composition $\tilde{\psi}^t$ of any order is also monotonically increasing. Furthermore, we can bound $\tilde{\psi}$ from below with

$$\tilde{\psi}(x) = (\eta - 1)|x| + \eta |x|^3 \geq \max\{(\eta - 1)|x|, \eta |x|^3\} \triangleq \varphi(x). \tag{42}$$

Thus,

$$|x_t| = |\psi^t(x_0)| \geq \varphi^t(x_0) \triangleq \chi_t. \tag{43}$$

Again, $\varphi : \mathbb{R}_+ \to \mathbb{R}_+$ is monotonically increasing, and therefore any composition with itself is also monotonically increasing on $\mathbb{R}_+$. Obviously,

$$\varphi(x) \geq (\eta - 1)|x| \qquad \text{and} \qquad \varphi(x) \geq \eta |x|^3. \tag{44}$$

Then we can bound $\chi_t$ by

$$\chi_t \geq (\eta - 1)^t |x_0|. \tag{45}$$

Since $\eta > 2$, there exists $T \in \mathbb{N}$ such that

$$\chi_T \geq (\eta - 1)^T |x_0| > 2. \tag{46}$$

Now, for all $t > T$ we have

$$\chi_t = \varphi^{t-T}(\chi_T). \tag{47}$$

Here, we will use the second bound, $i.e.$, $\varphi(x) \geq \eta |x|^3$, $t - T$ times as

$$\begin{aligned} \chi_t &= \underbrace{\eta |\eta \cdots \eta |\chi_T|^3 \cdots |^3|^3}_{t-T \text{ times}} \\ &= \eta^{(3^{t-T}-1)/2} |\chi_T|^{3^{t-T}} \\ &\geq 2^{(3^{t-T}-1)/2} 2^{3^{t-T}} \\ &= 2^{(3^{t-T+1}-1)/2}. \end{aligned} \tag{48}$$

Therefore, $\chi_t$ diverges with superlinear rate and so does $|x_t|$.

## D  CONDITION FOR STABLE OSCILLATIONS IN GD

In Sec. 5.1 we formulate the oscillations of GD as a flip bifurcation. We saw that the first Lyapunov coefficient $C_0$ controls the stability of the oscillations. For a general nonlinear map $\psi(x, \eta)$, this coefficient is given by (Kuznetsov, 1998, Sec. 5.4)

$$C_0 = \frac{1}{6} \langle u, \mathcal{D}_x^3 \psi(x^*, \eta)[v, v, v] \rangle - \frac{1}{2} \langle u, \mathcal{D}_x^2 \psi(x^*, \eta)[v, p] \rangle, \tag{49}$$

where $u$ and $v$ are normalized left and right eigenvectors of the Jacobian $\mathcal{D}_x \psi(x^*, \eta)$ corresponding to the eigenvalue $-1$, such that $\langle u, v \rangle = 1$, and

$$p = [\mathcal{D}_x \psi(x^*, \eta) - I]^{-1} \mathcal{D}_x^2 \psi(x^*, \eta)[v, v]. \tag{50}$$

We would like to write these expressions in terms of the loss function $\mathcal{L}$ for the GD dynamics. In this case, $\boldsymbol{\psi}(\boldsymbol{x}, \eta) = \boldsymbol{x} - \eta \nabla \mathcal{L}(\boldsymbol{x})$, and we have that the Jacobian is

$$\mathcal{D}_{\boldsymbol{x}} \boldsymbol{\psi}(\boldsymbol{x}^*, \eta) = \boldsymbol{I} - \eta \nabla^2 \mathcal{L}(\boldsymbol{x}^*). \tag{51}$$

Since this Jacobian is systematic, $\boldsymbol{v}$ equals $\boldsymbol{u}$. Moreover, it is easy to see that $\boldsymbol{v}$ is the top eigenvector of the Hessian, corresponding to $\lambda_{\max}(\nabla^2 \mathcal{L}(\boldsymbol{x}^*))$. Additionally,

$$\mathcal{D}_{\boldsymbol{x}}^2 \boldsymbol{\psi}(\boldsymbol{x}^*, \eta)[\boldsymbol{v}, \boldsymbol{v}] = -\eta \mathcal{D}^3 \mathcal{L}(\boldsymbol{x}^*)[\boldsymbol{v}, \boldsymbol{v}] = -\frac{\eta}{3} \nabla_{\boldsymbol{v}} \mathcal{D}^3 \mathcal{L}(\boldsymbol{x}^*)[\boldsymbol{v}, \boldsymbol{v}, \boldsymbol{v}]. \tag{52}$$

Thus,

$$\begin{aligned}
\boldsymbol{p} &= \left[ -\eta \nabla^2 \mathcal{L}(\boldsymbol{x}^*) \right]^{-1} \left( -\frac{\eta}{3} \nabla_{\boldsymbol{v}} \mathcal{D}^3 \mathcal{L}(\boldsymbol{x}^*)[\boldsymbol{v}, \boldsymbol{v}, \boldsymbol{v}] \right) \\
&= \frac{1}{3} \left[ \nabla^2 \mathcal{L}(\boldsymbol{x}^*) \right]^{-1} \nabla_{\boldsymbol{v}} \mathcal{D}^3 \mathcal{L}(\boldsymbol{x}^*)[\boldsymbol{v}, \boldsymbol{v}, \boldsymbol{v}] \\
&= \frac{1}{3} \boldsymbol{q}, \tag{53}
\end{aligned}$$

where

$$\boldsymbol{q} = \left[ \nabla^2 \mathcal{L}(\boldsymbol{x}^*) \right]^{-1} \nabla_{\boldsymbol{v}} \mathcal{D}^3 \mathcal{L}(\boldsymbol{x}^*)[\boldsymbol{v}, \boldsymbol{v}, \boldsymbol{v}]. \tag{54}$$

Next we have,

$$\left\langle \boldsymbol{u}, \mathcal{D}_{\boldsymbol{x}}^2 \boldsymbol{\psi}(\boldsymbol{x}^*, \eta)[\boldsymbol{v}, \boldsymbol{p}] \right\rangle = \frac{1}{3} \left\langle \boldsymbol{v}, \mathcal{D}_{\boldsymbol{x}}^2 \boldsymbol{\psi}(\boldsymbol{x}^*, \eta)[\boldsymbol{v}, \boldsymbol{q}] \right\rangle = -\frac{\eta}{3} \mathcal{D}^3 \mathcal{L}(\boldsymbol{x}^*)[\boldsymbol{v}, \boldsymbol{v}, \boldsymbol{q}]. \tag{55}$$

And the first term in $C_0$ is

$$\left\langle \boldsymbol{u}, \mathcal{D}_{\boldsymbol{x}}^3 \boldsymbol{\psi}(\boldsymbol{x}^*, \eta)[\boldsymbol{v}, \boldsymbol{v}, \boldsymbol{v}] \right\rangle = \left\langle \boldsymbol{v}, -\eta \mathcal{D}^4 \mathcal{L}(\boldsymbol{x}^*)[\boldsymbol{v}, \boldsymbol{v}, \boldsymbol{v}] \right\rangle = -\eta \mathcal{D}^4 \mathcal{L}(\boldsymbol{x}^*)[\boldsymbol{v}, \boldsymbol{v}, \boldsymbol{v}, \boldsymbol{v}]. \tag{56}$$

Overall,

$$C_0 = -\frac{\eta}{6} \mathcal{D}^4 \mathcal{L}(\boldsymbol{x}^*)[\boldsymbol{v}, \boldsymbol{v}, \boldsymbol{v}, \boldsymbol{v}] + \frac{\eta}{6} \mathcal{D}^3 \mathcal{L}(\boldsymbol{x}^*)[\boldsymbol{v}, \boldsymbol{v}, \boldsymbol{q}]. \tag{57}$$

A period-2 cycle near $\boldsymbol{x}^*$ is stable if and only if $C_0 > 0$ (Kuznetsov, 1998), which results in the condition

$$\mathcal{D}^3 \mathcal{L}(\boldsymbol{x}^*)[\boldsymbol{v}, \boldsymbol{v}, \boldsymbol{q}] > \mathcal{D}^4 \mathcal{L}(\boldsymbol{x}^*)[\boldsymbol{v}, \boldsymbol{v}, \boldsymbol{v}, \boldsymbol{v}]. \tag{58}$$

Originally, the scale of $\boldsymbol{v}$ was important for the magnitude of $C_0$. However, the scale of $\boldsymbol{v}$ has no effect on the sign of $C_0$, and thus has no impact on this condition.

## E   ANALYTIC EXAMPLE OF THEOREM 1

In this section we consider the oscillations of GD at the edge of stability on the function

$$f_\alpha(x) = \frac{1}{2} x^2 + \frac{\alpha}{6} x^3 + \frac{1}{8} x^4. \tag{59}$$

To apply Thm. 1, we denote $\mathcal{L} = f_\alpha$, and therefore

$$v = 1, \quad f_\alpha''(0) = 1, \quad f_\alpha^{(3)}(0) = \alpha, \quad f_\alpha^{(4)}(0) = 3. \tag{60}$$

Then,

$$q = \left[ f_\alpha''(0) \right]^{-1} \frac{d}{dv} \left( f_\alpha^{(3)}(0) v^3 \right) \Big|_{v=1} = 3\alpha v^2 \Big|_{v=1} = 3\alpha. \tag{61}$$

Overall,

$$\mathcal{D}^3 \mathcal{L}(x^*)[v, v, q] = f_\alpha^{(3)}(0) v^2 q = \alpha \cdot 3\alpha = 3\alpha^2. \tag{62}$$

$$\mathcal{D}^4 \mathcal{L}(x^*)[v, v, v, v] = f_\alpha^{(4)}(0) v^4 = 3. \tag{63}$$

Thus, the stability condition for oscillations is

$$\mathcal{D}^3 \mathcal{L}(x^*)[v, v, q] > \mathcal{D}^4 \mathcal{L}(x^*)[v, v, v, v] \quad \Longleftrightarrow \quad 3\alpha^2 > 3 \quad \Longleftrightarrow \quad |\alpha| > 1. \tag{64}$$

# F    PROOF OF THEOREM 2

Let $\{\mathcal{B}_i\}_{i=1}^N$ be all possible different batches of size $B$ form the dataset $\{f_i\}_{i=1}^n$, where $N = \binom{n}{B}$. Recall that $\hat{\psi}_{\mathcal{B}}$ denotes the of GD transform on batch $\mathcal{B}$, *i.e.*, taking a single gradient step with respect to $\hat{\mathcal{L}}_{\mathcal{B}}$ (see (19)). Moreover, let $\hat{\psi}_{\mathcal{B}}^t$ denote the application of $\hat{\psi}_{\mathcal{B}}$ for $t$ times. Namely,

$$\hat{\psi}_{\mathcal{B}}^t = \underbrace{\hat{\psi}_{\mathcal{B}} \circ \cdots \circ \hat{\psi}_{\mathcal{B}} \circ \hat{\psi}_{\mathcal{B}}}_{t \text{ times}}. \tag{65}$$

For a stochastic batch $\mathcal{B}_t$, $\hat{\psi}_{\mathcal{B}_t}$ is distributed uniformly over $\{\hat{\psi}_{\mathcal{B}_i}\}$, *i.e.*, for any $\boldsymbol{x} \in \mathbb{R}^d$

$$\hat{\psi}_{\mathcal{B}_i}(\boldsymbol{x}) \sim \mathcal{U}\left(\left\{\hat{\psi}_{\mathcal{B}_i}(\boldsymbol{x})\right\}_{i=1}^N\right). \tag{66}$$

Given an initial point $\boldsymbol{x}_0 \in \mathbb{R}^d$, assume that for some batch $\mathcal{B}_{i^*}$ (with index $i^*$) GD's iterates, denoted by $\{\boldsymbol{x}_t^{(\mathcal{B}_{i^*})}\}$, diverge with superlinear rate. That is

$$\left\|\boldsymbol{x}_t^{(\mathcal{B}_{i^*})} - \boldsymbol{x}^*\right\|^{\frac{1}{t}} \xrightarrow[t \to \infty]{} \infty, \tag{67}$$

Using our notation, we have $\boldsymbol{x}_t^{(\mathcal{B}_{i^*})} = \hat{\psi}_{\mathcal{B}_{i^*}}^t(\boldsymbol{x}_0)$. Let us look at the expectation of the distance between SGD iterates $\{\boldsymbol{x}_t\}$ form (10) and the minimizer $\boldsymbol{x}^*$

$$\mathbb{E}\left[\|\boldsymbol{x}_t - \boldsymbol{x}^*\|\right] = \frac{1}{N^t} \sum_{(i_1,i_2,\ldots,i_t) \in \{1,\ldots,N\}^t} \left\|\hat{\psi}_{\mathcal{B}_{i_t}} \circ \cdots \circ \hat{\psi}_{\mathcal{B}_{i_2}} \circ \hat{\psi}_{\mathcal{B}_{i_1}}(\boldsymbol{x}_0) - \boldsymbol{x}^*\right\|$$

$$\geq \frac{1}{N^t} \left\|\hat{\psi}_{\mathcal{B}_{i_t}} \circ \cdots \circ \hat{\psi}_{\mathcal{B}_{i_2}} \circ \hat{\psi}_{\mathcal{B}_{i_1}}(\boldsymbol{x}_0) - \boldsymbol{x}^*\right\|\Big|_{i_1 = i_2 = \cdots = i_t = i^*}$$

$$= \frac{1}{N^t} \left\|\hat{\psi}_{\mathcal{B}_{i^*}}^t(\boldsymbol{x}_0) - \boldsymbol{x}^*\right\|$$

$$= \exp\left\{\log\left(\left\|\hat{\psi}_{\mathcal{B}_{i^*}}^t(\boldsymbol{x}_0) - \boldsymbol{x}^*\right\|\right) - t\log(N)\right\}$$

$$= \exp\left\{t\left[\frac{1}{t}\log\left(\left\|\hat{\psi}_{\mathcal{B}_{i^*}}^t(\boldsymbol{x}_0) - \boldsymbol{x}^*\right\|\right) - \log(N)\right]\right\}$$

$$= \exp\left\{t\left[\log\left(\left\|\boldsymbol{x}_t^{(\mathcal{B}_{i^*})} - \boldsymbol{x}^*\right\|^{\frac{1}{t}}\right) - \log(N)\right]\right\} \xrightarrow[t \to \infty]{} \infty. \tag{68}$$

# G    INTRODUCTION TO SPECTRAL ANALYSIS OF LINEAR OPERATORS

Let us start with the following definitions.

**Definition 3 (Operator norm)** *Let $\boldsymbol{A}$ be a linear operator over a vector space $V$, then its operator norm is given by*

$$\|\boldsymbol{A}\| = \inf\{c \geq 0 \ : \ \|\boldsymbol{A}\boldsymbol{v}\| \leq c\|\boldsymbol{v}\| \quad \text{for all } \boldsymbol{v} \in V\}. \tag{69}$$

**Definition 4 (Spectrum)** *Let $\boldsymbol{A}$ be a linear operator over a Banach space $V$, then its spectrum is given by*

$$\sigma(\boldsymbol{A}) = \{\lambda \in \mathbb{C} \ : \ \boldsymbol{A} - \lambda\boldsymbol{I} \text{ is not bijective}\}, \tag{70}$$

*where $\boldsymbol{I}$ is the identity operator.*

**Definition 5 (Spectral radius)** *The spectral radius of an operator $\boldsymbol{A}$ is given by*

$$r(\boldsymbol{A}) = \sup_{\lambda \in \sigma(\boldsymbol{A})} |\lambda|. \tag{71}$$

Consider the following linear system

$$\boldsymbol{\mu}_{t+1} = \boldsymbol{A}\boldsymbol{\mu}_t, \tag{72}$$

where $\boldsymbol{A}$ is a bounded linear operator. We want to have some condition such that the iterates are bounded or converging. Here, we can unfold the equation to get an explicit formula for any $\boldsymbol{\mu}_t$ as

$$\boldsymbol{\mu}_t = \boldsymbol{A}^t \boldsymbol{\mu}_0. \tag{73}$$

A naive way to ensure convergence, $i.e.$, $\boldsymbol{\mu}_t \to \mathbf{0}$ as $t \to \infty$, is by taking the operator norm of $\boldsymbol{A}$ to be less than one, $i.e.$, $\|\boldsymbol{A}\| < 1$. Then

$$\|\boldsymbol{\mu}_t\| = \left\|\boldsymbol{A}^t\boldsymbol{\mu}_0\right\| \leq \|\boldsymbol{A}\|^t \|\boldsymbol{\mu}_0\| \to 0. \tag{74}$$

However, this is quite restrictive and will give us a loss condition. Note that we only like to know if $\|\boldsymbol{A}^t\|$ is bounded or shrinks to zero. In special cases, we can easily compute $\boldsymbol{A}^t$. For example, in the finite-dimensional case, where $\boldsymbol{A} = \boldsymbol{P}\boldsymbol{D}\boldsymbol{P}^{-1}$ is diagonalizable ($e.g.$, symmetric or normal). Then,

$$\boldsymbol{\mu}_t = \boldsymbol{A}^t\boldsymbol{\mu}_0 = \left(\boldsymbol{P}\boldsymbol{D}\boldsymbol{P}^{-1}\right)^t \boldsymbol{\mu}_0 = \boldsymbol{P}\boldsymbol{D}^t\boldsymbol{P}^{-1}\boldsymbol{\mu}_0. \tag{75}$$

Here, the system is stable if and only if the spectral radius of $\boldsymbol{A}$ is less or equal to one. If it is strictly less than one, then $\boldsymbol{\mu}_t \to \mathbf{0}$.

In the general case, we can use Gelfand's formula for bounded linear operators on Banach spaces. Let $r(\boldsymbol{A})$ denote the spectral radius of $\boldsymbol{A}$, then Gelfand's formula is

$$r(\boldsymbol{A}) = \lim_{t \to \infty} \left\|\boldsymbol{A}^t\right\|^{\frac{1}{t}} = \inf_{t \in \mathbb{N}} \left\|\boldsymbol{A}^t\right\|^{\frac{1}{t}}. \tag{76}$$

From this formula, we can see that if $r(\boldsymbol{A}) < 1$, then $\boldsymbol{\mu}_t \to \mathbf{0}$.

## H  COMPUTATION OF THE OPERATOR BLOCKS

In this section we give the missing steps from (24). To this end, denote

$$\boldsymbol{Y}_{t,k} = \frac{1}{k!}\mathcal{D}^k\hat{\psi}_{\mathcal{B}_t}(\boldsymbol{x}^*) \in \mathbb{R}^{d \times d^k}. \tag{77}$$

Then, the evolution over time of the $k$th moment is

$$\mathbb{E}\left[\left(\sum_{p=1}^{\infty} \frac{1}{p!}\mathcal{D}^p\hat{\psi}_{\mathcal{B}_t}(\boldsymbol{x}^*)\Delta\boldsymbol{x}_t^p\right)^{\otimes k}\right]$$

$$= \mathbb{E}\left[\left(\sum_{p=1}^{\infty}\boldsymbol{Y}_{t,p}\Delta\boldsymbol{x}_t^p\right)^{\otimes k}\right]$$

$$= \mathbb{E}\left[\sum_{p=k}^{\infty}\sum_{\substack{1 \leq \kappa_1,\cdots,\kappa_k \leq p-k+1 \\ \kappa_1+\kappa_2+\cdots+\kappa_k=p}}(\boldsymbol{Y}_{t,\kappa_1}\Delta\boldsymbol{x}_t^{\kappa_1}) \otimes (\boldsymbol{Y}_{t,\kappa_2}\Delta\boldsymbol{x}_t^{\kappa_2}) \otimes \cdots \otimes (\boldsymbol{Y}_{t,\kappa_k}\Delta\boldsymbol{x}_t^{\kappa_k})\right]$$

$$= \mathbb{E}\left[\sum_{p=k}^{\infty}\sum_{\substack{1 \leq \kappa_1,\cdots,\kappa_k \leq p-k+1 \\ \kappa_1+\kappa_2+\cdots+\kappa_k=p}}(\boldsymbol{Y}_{t,\kappa_1} \otimes \cdots \otimes \boldsymbol{Y}_{t,\kappa_k})(\Delta\boldsymbol{x}_t^{\kappa_1} \otimes \cdots \otimes \Delta\boldsymbol{x}_t^{\kappa_k})\right]$$

$$= \mathbb{E}\left[\sum_{p=k}^{\infty}\sum_{\substack{1 \leq \kappa_1,\cdots,\kappa_k \leq p-k+1 \\ \kappa_1+\kappa_2+\cdots+\kappa_k=p}}(\boldsymbol{Y}_{t,\kappa_1} \otimes \boldsymbol{Y}_{t,\kappa_2} \otimes \cdots \otimes \boldsymbol{Y}_{t,\kappa_k})\left(\Delta\boldsymbol{x}_t^{\sum_{i=1}^{k}\kappa_i}\right)\right]$$

$$= \mathbb{E}\left[\sum_{p=k}^{\infty}\sum_{\substack{1 \leq \kappa_1,\cdots,\kappa_k \leq p-k+1 \\ \kappa_1+\kappa_2+\cdots+\kappa_k=p}}(\boldsymbol{Y}_{t,\kappa_1} \otimes \boldsymbol{Y}_{t,\kappa_2} \otimes \cdots \otimes \boldsymbol{Y}_{t,\kappa_k})\Delta\boldsymbol{x}_t^p\right]$$

$$= \mathbb{E}\left[\sum_{p=k}^{\infty}\left(\sum_{\substack{1 \leq \kappa_1,\cdots,\kappa_k \leq p-k+1 \\ \kappa_1+\kappa_2+\cdots+\kappa_k=p}}\boldsymbol{Y}_{t,\kappa_1} \otimes \boldsymbol{Y}_{t,\kappa_2} \otimes \cdots \otimes \boldsymbol{Y}_{t,\kappa_k}\right)\Delta\boldsymbol{x}_t^p\right]$$

$$= \sum_{p=k}^{\infty} \mathbb{E} \left[ \sum_{\substack{1 \leq \kappa_1, \cdots, \kappa_k \leq p-k+1 \\ \kappa_1 + \kappa_2 + \cdots + \kappa_k = p}} \boldsymbol{Y}_{t,\kappa_1} \otimes \boldsymbol{Y}_{t,\kappa_2} \otimes \cdots \otimes \boldsymbol{Y}_{t,\kappa_k} \right] \mathbb{E} \left[ \Delta \boldsymbol{x}_t^p \right]$$

$$= \sum_{p=k}^{\infty} \boldsymbol{\Psi}_{k,p} \mathbb{E} \left[ \Delta \boldsymbol{x}_t^p \right], \tag{78}$$

where

$$\boldsymbol{\Psi}_{k,p} = \mathbb{E} \left[ \sum_{\substack{1 \leq \kappa_1, \cdots, \kappa_k \leq p-k+1 \\ \kappa_1 + \kappa_2 + \cdots + \kappa_k = p}} \boldsymbol{Y}_{t,\kappa_1} \otimes \boldsymbol{Y}_{t,\kappa_2} \otimes \cdots \otimes \boldsymbol{Y}_{t,\kappa_k} \right] \in \mathbb{R}^{d^k \times d^p}, \tag{79}$$

## I  BOUNDING THE OPERATOR

In this section, we assume that the condition of Thm. 3 holds. Then, we show that there exists a $\rho > 0$ such that $\boldsymbol{\Psi}_\rho$ is bounded. For this, we use the following result (see proof in App. L).

**Theorem 5** *Let $\boldsymbol{T}$ be an operator defined on $\ell_2$ space. Denote by $\{\boldsymbol{T}_{i,j}\}$ a division of $\boldsymbol{T}$ into blocks, such that $\forall i, j \; \boldsymbol{T}_{i,j} \in \mathbb{R}^{d_i \times d_j}$ where $\{d_i\}_{i=1}^{\infty}$ is a some sequence. Assume that*

$$\forall j \in \mathbb{N} \qquad \sum_{i=1}^{\infty} \|\boldsymbol{T}_{i,j}\| \leq \alpha \qquad \text{and} \qquad \forall i \in \mathbb{N} \qquad \sum_{j=1}^{\infty} \|\boldsymbol{T}_{i,j}\| \leq \beta. \tag{80}$$

*Then $\boldsymbol{T}$ is a bounded linear operator and*

$$\|\boldsymbol{T}\|_2 \leq \sqrt{\alpha \beta}. \tag{81}$$

Let us apply Thm. 5 to $\boldsymbol{\Psi}_\rho$. Using the definition of the blocks $\{\boldsymbol{\Psi}_{k,p}\}$ given in (79) we have

$$\|\boldsymbol{\Psi}_{k,p}\| = \left\| \mathbb{E} \left[ \sum_{\substack{1 \leq \kappa_1, \cdots, \kappa_k \leq p-k+1 \\ \kappa_1 + \kappa_2 + \cdots + \kappa_k = p}} \boldsymbol{Y}_{t,\kappa_1} \otimes \boldsymbol{Y}_{t,\kappa_2} \otimes \cdots \otimes \boldsymbol{Y}_{t,\kappa_k} \right] \right\|$$

$$\leq \mathbb{E} \left[ \sum_{\substack{1 \leq \kappa_1, \cdots, \kappa_k \leq p-k+1 \\ \kappa_1 + \kappa_2 + \cdots + \kappa_k = p}} \left\| \boldsymbol{Y}_{t,\kappa_1} \otimes \boldsymbol{Y}_{t,\kappa_2} \otimes \cdots \otimes \boldsymbol{Y}_{t,\kappa_k} \right\| \right]$$

$$= \mathbb{E} \left[ \sum_{\substack{1 \leq \kappa_1, \cdots, \kappa_k \leq p-k+1 \\ \kappa_1 + \kappa_2 + \cdots + \kappa_k = p}} \left\| \boldsymbol{Y}_{t,\kappa_1} \right\| \left\| \boldsymbol{Y}_{t,\kappa_2} \right\| \cdots \left\| \boldsymbol{Y}_{t,\kappa_k} \right\| \right], \tag{82}$$

where $\boldsymbol{Y}_{t,p}$ is given in (77). Let $\{\mathcal{B}_m\}_{m=1}^{N}$ be all possible different batches of size $B$ form the dataset $\{f_m\}_{m=1}^{n}$, where $N = \binom{n}{B}$. Since $\{f_m\}$ are analytic and $\{\hat{\mathcal{L}}_{\mathcal{B}_m}\}$ are finite sum losses, then also $\{\hat{\psi}_{\mathcal{B}_m}\}$ are analytic. Then, using Gevrey class theory, for each batch $\mathcal{B}_m$ there exists $C_m > 0$ such that

$$\max_{i,j} \left| \left[ \mathcal{D}^p \hat{\psi}_{\mathcal{B}_m}(\boldsymbol{x}^*) \right]_{i,j} \right| \leq C_m^{p+1} p! \qquad \forall p \geq 1, \tag{83}$$

where $[\mathcal{D}^p \hat{\psi}_{\mathcal{B}_m}(\boldsymbol{x}^*)]_{i,j}$ are the elements of in the matrix $\mathcal{D}^p \hat{\psi}_{\mathcal{B}_m}(\boldsymbol{x}^*)$, which are all the (mixed) partial derivatives of degree $p$. Setting

$$C = \max_{m \in [N]} C_m, \tag{84}$$

we get for a random batch $\mathcal{B}_t$

$$\max_{i,j} \left| \left[ \mathcal{D}^p \hat{\psi}_{\mathcal{B}_t}(\boldsymbol{x}^*) \right]_{i,j} \right| \leq C^{p+1} p! \quad \text{w.p. } 1 \qquad \forall p \geq 1. \tag{85}$$

Now that we have a uniform bound on all elements in the matrix, we can bound its norm. Specifically, it is well known that for a matrix $\boldsymbol{A} \in \mathbb{R}^{m \times n}$ with elements $|\boldsymbol{A}_{i,j}| \leq M$, we have $\|\boldsymbol{A}\| \leq M\sqrt{mn}$ (a simple application of Thm. 5 can give this result as well). Using this, and the fact that $\mathcal{D}^p \hat{\boldsymbol{\psi}}_{\mathcal{B}_t}(\boldsymbol{x}^*) \in \mathbb{R}^{d \times d^p}$ we get

$$\|\boldsymbol{Y}_{t,p}\| = \frac{1}{p!} \left\| \mathcal{D}^p \hat{\boldsymbol{\psi}}_{\mathcal{B}_t}(\boldsymbol{x}^*) \right\| \leq C^{p+1} d^{\frac{p+1}{2}} \quad \text{w.p. 1.} \tag{86}$$

Define

$$Q_{t,p} = \begin{cases} \|\boldsymbol{Y}_{t,1}\|, & p = 1, \\ C^{p+1} d^{\frac{p+1}{2}}, & \text{otherwise.} \end{cases} \tag{87}$$

Then for all $p \geq 1$ and $t \in \mathbb{N}$

$$\|\boldsymbol{Y}_{t,p}\| \leq Q_{t,p} \quad \text{w.p. 1,} \tag{88}$$

and

$$\|\boldsymbol{\Psi}_{k,p}\| \leq \mathbb{E}\left[ \sum_{\substack{1 \leq \kappa_1, \cdots, \kappa_k \leq p-k+1 \\ \kappa_1 + \kappa_2 + \cdots + \kappa_k = p}} Q_{t,\kappa_1} Q_{t,\kappa_2} \cdots Q_{t,\kappa_k} \right]. \tag{89}$$

Let us apply Thm. 5 on $\boldsymbol{\Psi}_\rho$, while assuming that $\rho < \frac{1}{C\sqrt{d}}$. For the sum of the row block we have

$$\sum_{p=k}^{\infty} \rho^{p-k} \|\boldsymbol{\Psi}_{k,p}\| \leq \sum_{p=k}^{\infty} \rho^{p-k} \mathbb{E}\left[ \sum_{\substack{1 \leq \kappa_1, \cdots, \kappa_k \leq p-k+1 \\ \kappa_1 + \kappa_2 + \cdots + \kappa_k = p}} Q_{t,\kappa_1} Q_{t,\kappa_2} \cdots Q_{t,\kappa_k} \right]$$

$$= \rho^{-k} \mathbb{E}\left[ \sum_{p=k}^{\infty} \left( \sum_{\substack{1 \leq \kappa_1, \cdots, \kappa_k \leq p-k+1 \\ \kappa_1 + \kappa_2 + \cdots + \kappa_k = p}} Q_{t,\kappa_1} Q_{t,\kappa_2} \cdots Q_{t,\kappa_k} \right) \rho^p \right]$$

$$= \rho^{-k} \mathbb{E}\left[ \left( \sum_{p=1}^{\infty} Q_{t,p} \rho^p \right)^k \right]$$

$$= \rho^{-k} \mathbb{E}\left[ \left( \|\boldsymbol{Y}_{t,1}\| \rho + \sum_{p=2}^{\infty} C^{p+1} d^{\frac{p+1}{2}} \rho^p \right)^k \right]$$

$$= \rho^{-k} \mathbb{E}\left[ \left( \|\boldsymbol{Y}_{t,1}\| \rho + C\sqrt{d} \sum_{p=2}^{\infty} \left( C\sqrt{d}\rho \right)^p \right)^k \right]$$

$$= \rho^{-k} \mathbb{E}\left[ \left( \|\boldsymbol{Y}_{t,1}\| \rho + C\sqrt{d} \frac{C^2 d \rho^2}{1 - C\sqrt{d}\rho} \right)^k \right]$$

$$= \mathbb{E}\left[ \left( \|\boldsymbol{Y}_{t,1}\| + \frac{C^3 d^{3/2} \rho}{1 - C\sqrt{d}\rho} \right)^k \right], \tag{90}$$

where in the sixth step we used

$$\sum_{p=2}^{\infty} q^p = \frac{q^2}{1-q}, \tag{91}$$

for $0 < q = C\sqrt{d}\rho < 1$. Here we assume the condition in (13) holds and that $\{\nabla^2 \hat{\mathcal{L}}_{\mathcal{B}_i}(\boldsymbol{x}^*)\}$ have full rank. This means that for every batch $\mathcal{B}_i$

$$0 < \eta \lambda_{\min}\left( \nabla^2 \hat{\mathcal{L}}_{\mathcal{B}_i}(\boldsymbol{x}^*) \right) < \eta \lambda_{\max}\left( \nabla^2 \hat{\mathcal{L}}_{\mathcal{B}_i}(\boldsymbol{x}^*) \right) < 2. \tag{92}$$

Recall that $\mathcal{D}\hat{\psi}_{\mathcal{B}_i}(\boldsymbol{x}^*) = \boldsymbol{I} - \eta\nabla^2\hat{\mathcal{L}}_{\mathcal{B}_i}(\boldsymbol{x}^*)$. Then it is easy to show that there exists $\varepsilon \in (0,1)$ such that

$$
\begin{aligned}
\max_{i\in[N]}\left\|\mathcal{D}\hat{\psi}_{\mathcal{B}_i}(\boldsymbol{x}^*)\right\| &= \max_{i\in[N]}\left\|\boldsymbol{I} - \eta\nabla^2\hat{\mathcal{L}}_{\mathcal{B}_i}(\boldsymbol{x}^*)\right\| \\
&= \max_{i\in[N]}\left\{\max\left\{1 - \eta\lambda_{\min}\left(\nabla^2\hat{\mathcal{L}}_{\mathcal{B}_i}(\boldsymbol{x}^*)\right), \eta\lambda_{\max}\left(\nabla^2\hat{\mathcal{L}}_{\mathcal{B}_i}(\boldsymbol{x}^*)\right) - 1\right\}\right\} \\
&= 1 - \varepsilon.
\end{aligned}
\tag{93}
$$

This meaning that

$$
\|\boldsymbol{Y}_{t,1}\| \leq \max_{i\in[N]}\left\|\mathcal{D}\hat{\psi}_{\mathcal{B}_i}(\boldsymbol{x}^*)\right\| = 1 - \varepsilon \quad \text{w.p. } 1.
\tag{94}
$$

Note that $\frac{C^3 d^{3/2}\rho}{1 - C\sqrt{d}\rho} \geq 0$, then we can further bound (90) by

$$
\mathbb{E}\left[\left(\|\boldsymbol{Y}_{t,1}\| + \frac{C^3 d^{3/2}\rho}{1 - C\sqrt{d}\rho}\right)^k\right] \leq \left(1 - \varepsilon + \frac{C^3 d^{3/2}\rho}{1 - C\sqrt{d}\rho}\right)^k.
\tag{95}
$$

In ordrer for this to be bounded for any $k \in \mathbb{N}$, we will require

$$
\begin{aligned}
&\quad 1 - \varepsilon + \frac{C^3 d^{3/2}\rho}{1 - C\sqrt{d}\rho} < 1 \\
&\Leftrightarrow \quad \frac{C^3 d^{3/2}\rho}{1 - C\sqrt{d}\rho} < \varepsilon \\
&\Leftrightarrow \quad C^3 d^{3/2}\rho + \varepsilon C\sqrt{d}\rho < \varepsilon \\
&\Leftrightarrow \quad \rho < \frac{\varepsilon}{C^3 d^{3/2} + \varepsilon C\sqrt{d}} \triangleq \rho^*.
\end{aligned}
\tag{96}
$$

Therefore, under the condition of $\rho < \rho^*$ there exists $\gamma \in (0,1)$ (for example, $\gamma = 1 - \varepsilon + \frac{C^3 d^{3/2}\rho}{1 - C\sqrt{d}\rho}$) such that

$$
\sum_{p=k}^{\infty}\rho^{p-k}\|\boldsymbol{\Psi}_{k,p}\| \leq \gamma^k.
\tag{97}
$$

This means that the rows sum is uniformly bounded. Now, for the column sums, under the same assumptions we get

$$
\begin{aligned}
\sup_p \sum_{k=1}^{p}\rho^{p-k}\|\boldsymbol{\Psi}_{k,p}\| &\leq \sum_{p=1}^{\infty}\sum_{k=1}^{p}\rho^{p-k}\|\boldsymbol{\Psi}_{k,p}\| \\
&= \sum_{k=1}^{\infty}\sum_{p=k}^{\infty}\rho^{p-k}\|\boldsymbol{\Psi}_{k,p}\| \\
&\leq \sum_{k=1}^{\infty}\gamma^k \\
&= \frac{\gamma}{1 - \gamma},
\end{aligned}
\tag{98}
$$

where the change of summation order in second step is justified since all elements are non-negative. Hence, under the same assumptions, the absolute columns sum is also uniformly bounded.

Overall, we see that the conditions of Thm. 3 are sufficient to find a neighborhood around the minimum, $\{\boldsymbol{x}_0 : \|\boldsymbol{x}_0 - \boldsymbol{x}^*\| < \rho\}$, such that the operator $\boldsymbol{\Psi}_\rho$ is bounded. *For completeness*, in App. J, we show that the condition in (13) is also necessary. Namely, if this condition is violated, $\boldsymbol{\Psi}_\rho$ is not bounded.

## J   NECESSARY CONDITION FOR BOUNDNESS

In this section we show that the condition in (13) is also necessary to bound the operator $\boldsymbol{\Psi}_\rho$. We bring this only to give a complete theoretical understanding, *yet we do not use this derivation to prove our results.*

For $\boldsymbol{\Psi}_\rho$ to be bounded, all of its submatrices must be bounded. Note that the diagonal blocks of this operator $\{\boldsymbol{\Psi}_{k,k}\}_{k=1}^\infty$ are independent of $\rho$. Therefore, we should have a condition, independent of $\rho$, for these submatrices to be bounded. These blocks are given by

$$\boldsymbol{\Psi}_{k,k} = \mathbb{E}\left[\left(\mathcal{D}\hat{\psi}_{\mathcal{B}_t}(\boldsymbol{x}^*)\right)^{\otimes k}\right]. \tag{99}$$

Note that

$$\mathcal{D}\hat{\psi}_{\mathcal{B}_t}(\boldsymbol{x}^*) = \mathcal{D}\left(\boldsymbol{x} - \eta\nabla\hat{\mathcal{L}}_{\mathcal{B}_t}(\boldsymbol{x})\right)\Big|_{\boldsymbol{x}=\boldsymbol{x}^*} = \boldsymbol{I} - \eta\nabla^2\hat{\mathcal{L}}_{\mathcal{B}_t}(\boldsymbol{x}^*). \tag{100}$$

For ease of reading and better interpretability, let

$$\boldsymbol{H}_\mathcal{B} \triangleq \nabla^2\hat{\mathcal{L}}_\mathcal{B}(\boldsymbol{x}^*) \tag{101}$$

denote the Hessian of the batch $\mathcal{B}$. Then

$$\boldsymbol{\Psi}_{k,k} = \mathbb{E}\left[\left(\boldsymbol{I} - \eta\boldsymbol{H}_{\mathcal{B}_t}\right)^{\otimes k}\right]. \tag{102}$$

Moreover, denote by $\boldsymbol{H}_{\max}$ the batch that has the largest maximal eigenvalue, that is

$$\boldsymbol{H}_{\max} = \arg\max_{\mathcal{B}:|\mathcal{B}|=B}\left\{\lambda_{\max}\left(\boldsymbol{H}_\mathcal{B}\right)\right\}, \tag{103}$$

and by $\boldsymbol{v}_{\max}$ its corresponding eigenvector (normalized). Note that $\boldsymbol{\Psi}_{k,k}$ is symmetric for all $k \in \mathbb{N}$, therefore

$$\|\boldsymbol{\Psi}_{k,k}\| = \max_{\boldsymbol{u}\in\mathbb{R}^{d^k}:\|\boldsymbol{u}\|=1}\left|\boldsymbol{u}^\mathrm{T}\boldsymbol{\Psi}_{k,k}\boldsymbol{u}\right|. \tag{104}$$

Since $\|\boldsymbol{v}_{\max}^{\otimes k}\| = \|\boldsymbol{v}_{\max}\|^k = 1$, we have that

$$\begin{aligned}
\|\boldsymbol{\Psi}_{k,k}\| &\geq \left|\left(\boldsymbol{v}_{\max}^{\otimes k}\right)^\mathrm{T}\boldsymbol{\Psi}_{k,k}\boldsymbol{v}_{\max}^{\otimes k}\right| \\
&= \left|\left(\boldsymbol{v}_{\max}^{\otimes k}\right)^\mathrm{T}\mathbb{E}\left[\left(\boldsymbol{I} - \eta\boldsymbol{H}_{\mathcal{B}_t}\right)^{\otimes k}\right]\boldsymbol{v}_{\max}^{\otimes k}\right| \\
&= \left|\mathbb{E}\left[\left(\boldsymbol{v}_{\max}^{\otimes k}\right)^\mathrm{T}\left(\boldsymbol{I} - \eta\boldsymbol{H}_{\mathcal{B}_t}\right)^{\otimes k}\boldsymbol{v}_{\max}^{\otimes k}\right]\right| \\
&= \left|\mathbb{E}\left[\left(1 - \eta\boldsymbol{v}_{\max}^\mathrm{T}\boldsymbol{H}_{\mathcal{B}_t}\boldsymbol{v}_{\max}\right)^k\right]\right|. 
\end{aligned} \tag{105}$$

Assume that

$$\eta > \frac{2}{\lambda_{\max}(\boldsymbol{H}_{\max})}. \tag{106}$$

Since $\lambda_{\max}(\boldsymbol{H}_{\max}) = \boldsymbol{v}_{\max}^\mathrm{T}\boldsymbol{H}_{\max}\boldsymbol{v}_{\max}$, under the assumption above, we have that

$$\mathbb{P}\left(\eta\boldsymbol{v}_{\max}^\mathrm{T}\boldsymbol{H}_{\mathcal{B}_t}\boldsymbol{v}_{\max} > 2\right) > 0. \tag{107}$$

Therefore, continuing from (105)

$$\begin{aligned}
\|\boldsymbol{\Psi}_{k,k}\| &\geq \left|\mathbb{E}\left[\left(1 - \eta\boldsymbol{v}_{\max}^\mathrm{T}\boldsymbol{H}_{\mathcal{B}_t}\boldsymbol{v}_{\max}\right)^k\right]\right| \\
&= \left|\mathbb{P}\left(\eta\boldsymbol{v}_{\max}^\mathrm{T}\boldsymbol{H}_{\mathcal{B}_t}\boldsymbol{v}_{\max} > 2\right)\mathbb{E}\left[\left(1 - \eta\boldsymbol{v}_{\max}^\mathrm{T}\boldsymbol{H}_{\mathcal{B}_t}\boldsymbol{v}_{\max}\right)^k\Big|\eta\boldsymbol{v}_{\max}^\mathrm{T}\boldsymbol{H}_{\mathcal{B}_t}\boldsymbol{v}_{\max} > 2\right]\right. \\
&\quad \left. + \mathbb{P}\left(\eta\boldsymbol{v}_{\max}^\mathrm{T}\boldsymbol{H}_{\mathcal{B}_t}\boldsymbol{v}_{\max} \leq 2\right)\mathbb{E}\left[\left(1 - \eta\boldsymbol{v}_{\max}^\mathrm{T}\boldsymbol{H}_{\mathcal{B}_t}\boldsymbol{v}_{\max}\right)^k\Big|\eta\boldsymbol{v}_{\max}^\mathrm{T}\boldsymbol{H}_{\mathcal{B}_t}\boldsymbol{v}_{\max} \leq 2\right]\right| \\
&\geq \mathbb{P}\left(\eta\boldsymbol{v}_{\max}^\mathrm{T}\boldsymbol{H}_{\mathcal{B}_t}\boldsymbol{v}_{\max} > 2\right)\mathbb{E}\left[\left(\eta\boldsymbol{v}_{\max}^\mathrm{T}\boldsymbol{H}_{\mathcal{B}_t}\boldsymbol{v}_{\max} - 1\right)^k\Big|\eta\boldsymbol{v}_{\max}^\mathrm{T}\boldsymbol{H}_{\mathcal{B}_t}\boldsymbol{v}_{\max} > 2\right] \\
&\quad - \mathbb{P}\left(\eta\boldsymbol{v}_{\max}^\mathrm{T}\boldsymbol{H}_{\mathcal{B}_t}\boldsymbol{v}_{\max} \leq 2\right)\left|\mathbb{E}\left[\left(1 - \eta\boldsymbol{v}_{\max}^\mathrm{T}\boldsymbol{H}_{\mathcal{B}_t}\boldsymbol{v}_{\max}\right)^k\Big|\eta\boldsymbol{v}_{\max}^\mathrm{T}\boldsymbol{H}_{\mathcal{B}_t}\boldsymbol{v}_{\max} \leq 2\right]\right|,
\end{aligned} \tag{108}$$

where in the second step we used the law of total expectation, and in last step we used the triangle in-equality. Since $\boldsymbol{x}^*$ is an interpolating minimum, then $\boldsymbol{H}_{\mathcal{B}_t}$ is PSD w.p. one, and $0 \leq \boldsymbol{v}_{\max}^{\mathrm{T}} \boldsymbol{H}_{\mathcal{B}_t} \boldsymbol{v}_{\max}$. Thus,

$$\left| \mathbb{E}\left[ \left(1 - \eta \boldsymbol{v}_{\max}^{\mathrm{T}} \boldsymbol{H}_{\mathcal{B}_t} \boldsymbol{v}_{\max}\right)^k \middle| \eta \boldsymbol{v}_{\max}^{\mathrm{T}} \boldsymbol{H}_{\mathcal{B}_t} \boldsymbol{v}_{\max} \leq 2\right] \right| \leq 1. \tag{109}$$

However,

$$\mathbb{E}\left[ \left(\eta \boldsymbol{v}_{\max}^{\mathrm{T}} \boldsymbol{H}_{\mathcal{B}_t} \boldsymbol{v}_{\max} - 1\right)^k \middle| \eta \boldsymbol{v}_{\max}^{\mathrm{T}} \boldsymbol{H}_{\mathcal{B}_t} \boldsymbol{v}_{\max} > 2\right] \underset{k \to \infty}{\longrightarrow} \infty. \tag{110}$$

This means that under the condition in (106), we have that $\{\boldsymbol{\Psi}_{k,k}\}$ are unbounded. Therefore, a necessary condition for boundness is

$$\eta \leq \frac{2}{\lambda_{\max}(\boldsymbol{H}_{\max})}. \tag{111}$$

## K  SPECTRAL ANALYSIS

In App.I we proved that, under the condition in (13) of Thm. 3, we can find a neighborhood $\|\boldsymbol{x}_0 - \boldsymbol{x}^*\| < \rho$ such that the operator $\boldsymbol{\Psi}_\rho$ is bounded. In this section we show that under the same condition, the spectral radius of $\boldsymbol{\Psi}_\rho$, denoted by $r(\boldsymbol{\Psi}_\rho)$, is less than one. To do this, we first show that the operator is compact, which means that all the non-zero elements in its spectrum are eigenvalues (point spectrum). For this end, we define the following sequence of finite rank approximations (truncations) $\{\boldsymbol{\Psi}_\rho^k\}$, comprised of the first $k \times k$ blocks of $\boldsymbol{\Psi}_\rho$. Namely,

$$\boldsymbol{\Psi}_\rho^k = \begin{bmatrix} \boldsymbol{\Psi}_{1,1} & \rho\boldsymbol{\Psi}_{1,2} & \rho^2\boldsymbol{\Psi}_{1,3} & \cdots & \rho^{k-1}\boldsymbol{\Psi}_{1,k} & \boldsymbol{0} & \cdots \\ \boldsymbol{0} & \boldsymbol{\Psi}_{2,2} & \rho\boldsymbol{\Psi}_{2,3} & \cdots & \rho^{k-2}\boldsymbol{\Psi}_{2,k} & \boldsymbol{0} & \cdots \\ \boldsymbol{0} & \boldsymbol{0} & \boldsymbol{\Psi}_{3,3} & \cdots & \rho^{k-3}\boldsymbol{\Psi}_{3,k} & \boldsymbol{0} & \cdots \\ \vdots & \vdots & \vdots & \ddots & \vdots & \boldsymbol{0} & \cdots \\ \boldsymbol{0} & \boldsymbol{0} & \boldsymbol{0} & \boldsymbol{0} & \boldsymbol{\Psi}_{k,k} & \boldsymbol{0} & \cdots \\ \boldsymbol{0} & \boldsymbol{0} & \boldsymbol{0} & \boldsymbol{0} & \boldsymbol{0} & \boldsymbol{0} & \cdots \\ \vdots & \vdots & \vdots & \vdots & \vdots & \vdots & \ddots \end{bmatrix}. \tag{112}$$

Furthermore, define $\tilde{\boldsymbol{\Psi}}_{i,j}$ as the embedding of the block $\boldsymbol{\Psi}_{i,j}$ to the full space, *i.e.*

$$\tilde{\boldsymbol{\Psi}}_{i,j} = \begin{bmatrix} \boldsymbol{0} & \cdots & \boldsymbol{0} & \boldsymbol{0} & \boldsymbol{0} & \cdots \\ \vdots & \ddots & \vdots & \vdots & \vdots & \cdots \\ \boldsymbol{0} & \cdots & \boldsymbol{0} & \boldsymbol{0} & \boldsymbol{0} & \cdots \\ \boldsymbol{0} & \cdots & \boldsymbol{0} & \boldsymbol{\Psi}_{i,j} & \boldsymbol{0} & \cdots \\ \boldsymbol{0} & \cdots & \boldsymbol{0} & \boldsymbol{0} & \boldsymbol{0} & \cdots \\ \vdots & \vdots & \vdots & \vdots & \vdots & \ddots \end{bmatrix}, \tag{113}$$

such that

$$\boldsymbol{\Psi}_\rho = \sum_{i=1}^{\infty} \sum_{j=i}^{\infty} \rho^{j-i} \tilde{\boldsymbol{\Psi}}_{i,j} \qquad \text{and} \qquad \boldsymbol{\Psi}_\rho^k = \sum_{i=1}^{k} \sum_{j=i}^{k} \rho^{j-i} \tilde{\boldsymbol{\Psi}}_{i,j}. \tag{114}$$

Then,

$$\begin{aligned} \left\| \boldsymbol{\Psi}_\rho - \boldsymbol{\Psi}_\rho^k \right\| &= \left\| \sum_{i=1}^{\infty} \sum_{j=i}^{\infty} \rho^{j-i} \tilde{\boldsymbol{\Psi}}_{i,j} - \sum_{i=1}^{k} \sum_{j=i}^{k} \rho^{j-i} \tilde{\boldsymbol{\Psi}}_{i,j} \right\| \\ &= \left\| \sum_{i=k+1}^{\infty} \sum_{j=i}^{\infty} \rho^{j-i} \tilde{\boldsymbol{\Psi}}_{i,j} + \sum_{i=1}^{k} \sum_{j=k+1}^{\infty} \rho^{j-i} \tilde{\boldsymbol{\Psi}}_{i,j} \right\| \\ &\leq \sum_{i=k+1}^{\infty} \sum_{j=i}^{\infty} \rho^{j-i} \left\| \tilde{\boldsymbol{\Psi}}_{i,j} \right\| + \sum_{i=1}^{k} \sum_{j=k+1}^{\infty} \rho^{j-i} \left\| \tilde{\boldsymbol{\Psi}}_{i,j} \right\| \end{aligned}$$

$$= \sum_{i=k+1}^{\infty} \sum_{j=i}^{\infty} \rho^{j-i} \left\| \boldsymbol{\Psi}_{i,j} \right\| + \sum_{i=1}^{k} \sum_{j=k+1}^{\infty} \rho^{j-i} \left\| \boldsymbol{\Psi}_{i,j} \right\|$$

$$= \sum_{i=1}^{\infty} \sum_{j=i}^{\infty} \rho^{j-i} \left\| \boldsymbol{\Psi}_{i,j} \right\| - \sum_{i=1}^{k} \sum_{j=1}^{k} \rho^{j-i} \left\| \boldsymbol{\Psi}_{i,j} \right\| \xrightarrow[k \to \infty]{} 0, \tag{115}$$

where in the third step we used the fact that

$$\sum_{i=1}^{\infty} \sum_{j=i}^{\infty} \rho^{j-i} \left\| \tilde{\boldsymbol{\Psi}}_{i,j} \right\| = \sum_{i=1}^{\infty} \sum_{j=i}^{\infty} \rho^{j-i} \left\| \boldsymbol{\Psi}_{i,j} \right\| \tag{116}$$

is bounded (see (98)). Therefore, $\boldsymbol{\Psi}_\rho^k \xrightarrow{\|\cdot\|} \boldsymbol{\Psi}_\rho$ as $k \to \infty$, and thus $\boldsymbol{\Psi}_\rho$ is compact. This means that the non-zero elements in the spectrum of $\boldsymbol{\Psi}_\rho$ are comprised of its eigenvalues only (point spectrum).

In the following we use a known result about the convergence of the spectrum of finite rank approximations.

**Lemma 1 (Dunford & Schwartz (1964, Cp. XI.9 Lemma 5))** *Let $\{\boldsymbol{T}_k\}$ and $\boldsymbol{T}$ be compact operators, such that $\boldsymbol{T}_k \xrightarrow{\|\cdot\|} \boldsymbol{T}$. Let $\lambda_m(\boldsymbol{T})$ be an enumeration of the non-zero eigenvalues of $\boldsymbol{T}$, each repeated according to its multiplicity. Then there exist enumerations $\lambda_m(\boldsymbol{T}_k)$ of the non-zero eigenvalues of $\{\boldsymbol{T}_k\}$, with the repetitions according to multiplicity, such that*

$$\lim_{k \to \infty} \lambda_m(\boldsymbol{T}_k) = \lambda_m(\boldsymbol{T}), \qquad m \geq 1, \tag{117}$$

*where the limit is uniform in $m$.*

Let $\sigma(\cdot)$ denote the spectrum of an operator. Here, each $\boldsymbol{\Psi}_\rho^k$, when restricted to its square support, is a block upper triangular matrix. Hence, its spectrum[3] is given by the union of the eigenvalues of the blocks in the diagonal. Namely,

$$\sigma\left(\boldsymbol{\Psi}_\rho^k\right) = \bigcup_{j=1}^{k} \sigma\left(\boldsymbol{\Psi}_{j,j}\right). \tag{118}$$

Thus, according to Lemma 1 we have that the non-zero spectrum of $\boldsymbol{\Psi}_\rho$ is

$$\sigma\left(\boldsymbol{\Psi}_\rho\right) \setminus \{0\} = \lim_{k \to \infty} \sigma\left(\boldsymbol{\Psi}_\rho^k\right) \setminus \{0\} = \bigcup_{k=1}^{\infty} \sigma\left(\boldsymbol{\Psi}_{k,k}\right) \setminus \{0\}. \tag{119}$$

Now that we have the spectrum of $\boldsymbol{\Psi}_\rho$ we turn to show that under the condition of Thm. 3 in (13), the spectral radius $r(\boldsymbol{\Psi}_\rho)$ is less than one. Due to (119), it is sufficient to show that for all $k \in \mathbb{N}$ we have $r(\boldsymbol{\Psi}_{k,k}) \leq c < 1$, for some constant $c \in (0,1)$. Recall that

$$\boldsymbol{\Psi}_{k,k} = \mathbb{E}\left[\left(\mathcal{D}\hat{\psi}_{\mathcal{B}_t}(\boldsymbol{x}^*)\right)^{\otimes k}\right], \tag{120}$$

where $\mathcal{D}\hat{\psi}_{\mathcal{B}_t}(\boldsymbol{x}^*) = \boldsymbol{I} - \eta \nabla^2 \hat{\mathcal{L}}_{\mathcal{B}_t}(\boldsymbol{x}^*)$ is symmetric. Therefore, $\boldsymbol{\Psi}_{k,k}$ is also symmetric, and we have that $r(\boldsymbol{\Psi}_{k,k}) = \|\boldsymbol{\Psi}_{k,k}\|$. Thus, using Jensen's inequality

$$r\left(\boldsymbol{\Psi}_{k,k}\right) = \left\| \boldsymbol{\Psi}_{k,k} \right\|$$

$$= \left\| \mathbb{E}\left[\left(\mathcal{D}\hat{\psi}_{\mathcal{B}_t}(\boldsymbol{x}^*)\right)^{\otimes k}\right] \right\|$$

$$\leq \mathbb{E}\left[\left\| \left(\mathcal{D}\hat{\psi}_{\mathcal{B}_t}(\boldsymbol{x}^*)\right)^{\otimes k} \right\|\right]$$

$$= \mathbb{E}\left[\left\| \mathcal{D}\hat{\psi}_{\mathcal{B}_t}(\boldsymbol{x}^*) \right\|^k\right]. \tag{121}$$

---

[3]Without the zero eigenvalue.

Note that under the conditions of Thm. 3 we have $\left\| \mathcal{D}\hat{\psi}_{\mathcal{B}_t}(\boldsymbol{x}^*) \right\| \leq 1 - \varepsilon$ w.p. 1 for some $\varepsilon \in (0, 1)$ (see (94), and the discussion above it). Therefore,

$$r\left(\boldsymbol{\Psi}_{k,k}\right) \leq \mathbb{E}\left[\left\| \mathcal{D}\hat{\psi}_{\mathcal{B}_t}(\boldsymbol{x}^*) \right\|^k\right] \leq (1 - \varepsilon)^k. \tag{122}$$

Overall,

$$r\left(\boldsymbol{\Psi}_\rho\right) = \sup_{k \in \mathbb{N}} r\left(\boldsymbol{\Psi}_{k,k}\right) \leq \sup_{k \in \mathbb{N}} (1 - \varepsilon)^k = 1 - \varepsilon < 1. \tag{123}$$

## L  PROOF OF THEOREM 5

Let $\boldsymbol{T}$ be an operator defined on $\ell_2$ space. Assume that $\boldsymbol{T}$ consists of blocks $\{\boldsymbol{T}_{i,j}\}$, such that $\forall i, j\ \boldsymbol{T}_{i,j} \in \mathbb{R}^{d_i \times d_j}$ where $\{d_i\}_{i=1}^\infty$ is a given sequence. Additionally, assume that

$$\forall j \in \mathbb{N} \qquad \sum_{i=1}^\infty \|\boldsymbol{T}_{i,j}\| \leq \alpha \qquad \text{and} \qquad \forall i \in \mathbb{N} \qquad \sum_{j=1}^\infty \|\boldsymbol{T}_{i,j}\| \leq \beta. \tag{124}$$

Furthermore, for any $\boldsymbol{u} \in \ell_2$, denote by $\boldsymbol{u}_i$ its $i$th segment, such that $\boldsymbol{u}_i \in \mathbb{R}^{d_i}$. Then we have

$$
\begin{aligned}
\|\boldsymbol{T}\boldsymbol{u}\|^2 &= \sum_{i=1}^\infty \left\| \sum_{j=1}^\infty \boldsymbol{T}_{i,j}\boldsymbol{u}_j \right\|^2 \\
&\leq \sum_{i=1}^\infty \left( \sum_{j=1}^\infty \|\boldsymbol{T}_{i,j}\| \|\boldsymbol{u}_j\| \right)^2 \\
&= \sum_{i=1}^\infty \left( \sum_{j=1}^\infty \sqrt{\|\boldsymbol{T}_{i,j}\|} \sqrt{\|\boldsymbol{T}_{i,j}\|} \|\boldsymbol{u}_j\| \right)^2 \\
&\leq \sum_{i=1}^\infty \left( \sum_{j=1}^\infty \|\boldsymbol{T}_{i,j}\| \right) \left( \sum_{j=1}^\infty \|\boldsymbol{T}_{i,j}\| \|\boldsymbol{u}_j\|^2 \right) \\
&\leq \sum_{i=1}^\infty \beta \left( \sum_{j=1}^\infty \|\boldsymbol{T}_{i,j}\| \|\boldsymbol{u}_j\|^2 \right) \\
&= \beta \sum_{j=1}^\infty \|\boldsymbol{u}_j\|^2 \sum_{i=1}^\infty \|\boldsymbol{T}_{i,j}\| \\
&\leq \beta \sum_{j=1}^\infty \|\boldsymbol{u}_j\|^2 \alpha \\
&= \alpha\beta\|\boldsymbol{u}\|^2,
\end{aligned}
\tag{125}
$$

where in the second step we used the triangle inequality, the fourth step is due to Cauchy-Schwarz inequality, and in the sixth step we used the fact that all summands are non-negative, and therefore we can change summation order.

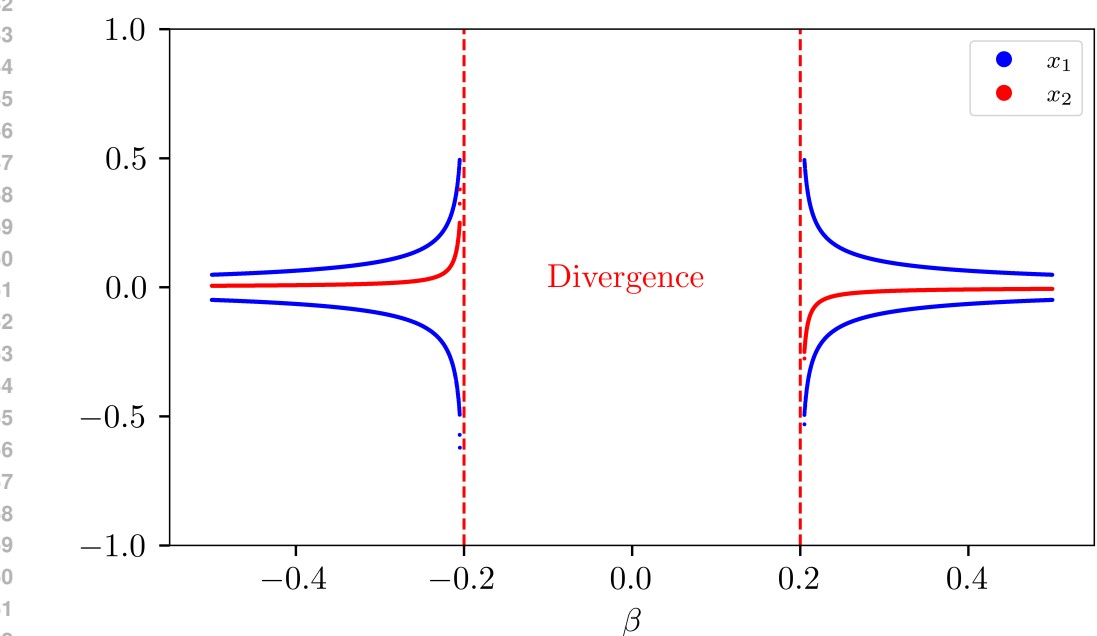

Figure 3: The final points of GD (last iteration) when applied to $f_\beta$. This is the example from the rebuttal.

## M   ANALYTIC EXAMPLE FROM THE REBUTTAL

In the rebuttal, we introduced the function $f_\beta : \mathbb{R}^2 \to \mathbb{R}$

$$f_\beta(\boldsymbol{x}) = \frac{1}{2}x_1^2 + \frac{1}{10}x_2^2 + \beta x_1^2 x_2 + \frac{1}{10}x_1^4. \tag{126}$$

We saw that the condition for a stable period-2 cycle in this case is $|\beta| > 0.2$. Here we present a simulation for this example, which illustrates that our condition correctly captures the stability of the period-2 cycle of GD.

