# OpenReview forum: "On the Stability of Nonlinear Dynamics in GD and SGD: Beyond Quadratic Potentials"
_ICLR.cc/2026/Conference — Submitted to ICLR 2026_

### Official Review · Reviewer_gWeu · 2025-10-28

**Soundness:** 3
**Presentation:** 3
**Contribution:** 3
**Rating:** 6
**Confidence:** 4

**Summary:**

This paper studies the training dynamics of gradient descent and stochastic gradient descent of analytic functions. The authors use dynamical system techniques to characterize the potential oscillations of GD and stability of SGD under interpolation regime.

Specifically, in Section 3 (GD), the authors provide in Theorem 1 a sufficient and necessary condition for the existence of a period-2 cycle near a local minimizer. Then in Section 4 (SGD), the authors showcase conditions (Theorem 2 and 3) under which the SGD iterates will diverge or converge to an interpolating minimizer. Section 5 gives a brief discussion on the connections between period-2 cycle and bifurcation, as well as a sketch of proof for SGD results.

The authors finally provide some related work, limitations and future work.

**Strengths:**

This paper gives a solid study of GD and SGD training dynamics on analytic functions. The authors leverage mathematical tools in dynamical system to show the nonlinear dynamics such as existence of periodic cycles.

**Weaknesses:**

1. Theory: despite the solid study of nonlinear dynamics of GD and SGD, there are some limitations in the current theoretical results:

1.1 In Section 3 (GD), it seems there are only results on the existence of period-2 cycles. Some phenomena, such as chaos, seem to be ignored.

1.2. In Section 4 (SGD), it seems only the interpolating minimizers are discussed, and the results only show the stability of SGD without any behaviors such as periodic cycles or chaos.

It would be good if the authors could provide some discussions on the difficulties of obtaining such missing results.

2. Experiments: there are only some simulations of analytic functions in this paper. It would be better if the authors could provide some real-world examples, in which GD has periodic behaviors in Section 3, or SGD with interpolation regime has the stability properties in Section 4.

3. Minor: the main audience of ICLR might not be familiar with some math concepts such as analytic functions. The authors might want to provide some brief definition/notation sections.

**Questions:**

I'm wondering if the authors could provide some comments and discussions on the weakness part mentioned above.

---

> ### Author Response · Authors · 2025-11-20
> **Rebuttal by Authors**
>
> Thanks for your review.
>
> **Update on prior work:**\
> Other reviewers brought to our attention prior work by Chen & Bruna (2022), which analyzed the stability of period-2 cycles in GD. We provide a detailed comparison above. In brief, they present a rigorous analysis of the univariate case and derive the condition for stable oscillations. However, as we discuss in detail above, it was unclear to them how this condition generalizes to the multivariate setting, highlighting that our multivariate result is nontrivial.
>
>
> **In Section 3 (GD), it seems there are only results on the existence of period-2 cycles. Some phenomena, such as chaos, seem to be ignored:**\
> Our analysis focuses on the typical case where the dynamics operate at the edge of stability, which corresponds to a period-2 cycle. Although higher-order cycles (such as period-4) can in principle occur, they appear to be less likely. At present, we do not have the mathematical tools to prove the existence of stable higher-period oscillations in high dimensions. Nevertheless, the condition in Theorem 1 would remain a necessary condition for any such oscillations.
>
> **In Section 4 (SGD), it seems only the interpolating minimizers are discussed:**\
> Our analysis for SGD is restricted to interpolating minima. This choice stems from the difficulty of defining stability of the full dynamics with respect to non-interpolating minima. In such cases, SGD cannot converge exactly to the minimizer; even if the dynamics are stable, the algorithm exhibits an inherent bias in expectation (Défossez & Bach, 2015). As a result, it is unclear whether the final distance from the minimizer reflects this bias or whether the algorithm converges to a different minimum altogether. In this work, we adopt the convergence-in-expectation perspective, but we believe that developing a more principled definition of stability for the non-interpolating case remains an important direction for future research.
>
> **In Section 4 (SGD), the results only show the stability of SGD without any behaviors such as periodic cycles or chaos:**\
> We may be misunderstanding what you mean by periodic cycles in the context of SGD. Since the iterates are stochastic, they usually don’t form truly periodic patterns.
>
> **Experiments:**\
> We will consider adding experiments on real-world applications to illustrate our theoretical findings. We would just like to note, however, that our conditions involve higher-order derivatives, which may be challenging to compute in practice.
>
> **The main audience of ICLR might not be familiar with some math concepts:**\
> We will add a section describing the definitions and notations in the paper. Thanks!

---

### Official Review · Reviewer_uH3Y · 2025-11-01

**Soundness:** 3
**Presentation:** 2
**Contribution:** 1
**Rating:** 0
**Confidence:** 5

**Summary:**

The paper studies stability of gradient descent (GD) and stochastic gradient descent (SGD) beyond linearization. The central claims are:
- GD can exhibit period 2 orbits even if the dynamical system is non-linear.
- For SGD stability may not depend on an average quantity but on upper-lowe bounds.
I find both claims either trivial or anyways not novel given the literature both in dynamical systems or on EoS in the ML community.
It is wellknown that nonlinear effects qualitatively change the stability picture, the problem has been addressed already but the authors failed to deal with the previous papers addressing them.

**Strengths:**

The paper has good examples and gives ideas cleanly.

**Weaknesses:**

### **Previous Work on Stability of SGD**

There are a few papers, I believe from Wu et al (2023) and Andreyev and Beneventano (2025) which should be discussed as direct competitor. The former discussing interpolating minima, the latter one being empirical and discussing the fact that more notions of stability are present and exactly picking one that seems to explain the trajectory in neural networks.
I understand this paper is not about neural networks but the significance of it is for neural networks otherwise would not be submitted to this conference. It needs to discuss the related literature in detail.

### **The claims for SGD**
It is wellknown that the necessary condition may be about the worst batch in SGD, previous work already tries to push further, thus this article not only is not novel but it is saying something trivial in the case of SGD. In particular, the results are implicit in the search for *a quantity* of previous works, which do establish that the *correct quantity* will be an expected quantity.

### **The claims for GD**
The claims for GD are not novel, they are present both in the dynamical systems and in the ML theory, see, e.g., Chen, Bruna et al (2023).
The result of Cohen et al. (2021) can be seen as a surprising statement that even thought the system is nonlinear, it acts as a linear one approximately. This is not addressed, and this is a central point of the old paper on EoS. It is thus not only not surprising, but it is wellknown and one of the reasons why EoS was surprising is that this result is wellknown.

Even assuming this is not known to some of us, there is substantial overlap between the paper’s GD contributions and prior work by Chen & Bruna on unstable convergence and period‑2 orbits of GD beyond the edge of stability. In Beyond the Edge of Stability via Two‑step Gradient Updates (arXiv 2022; ICML 2023), Chen & Bruna they:
- Demonstrate existence of stable period‑2 orbits for GD when the stepsize exceeds the linear stability limit, with a local condition involving (derivatives up to) third order guaranteeing convergence of a two‑step map to a period‑2 fixed point. They analyze canonical nonconvex settings and give intuition/observations for higher‑dimensional problems (matrix factorization) where period‑2 oscillations appear and can lead to further period‑doubling/chaos.
- Conceptually and empirically, they show GD can converge via a 2‑cycle even though the fixed point is linearly unstable—precisely the phenomenon highlighted in Sec. 2.1 and Thm. 1 of this submission (see Fig. 1–2 and discussion, pp. 3–5).

Thus the claims of the article are **extremely incremental, if not encompassed by literature published at a similar venue 3 years ago.**

### **There are no experiments**
Do you have any experiments on NN to substantiate your claims, you would see that EoS for stochastic setting **is** a phenomenon *on average*. On top of this, you would see that your necessary and sufficient bounds are so suboptimal to be meaningless

**Questions:**

Please comment on the weaknesses above.

---

> ### Author Response · Authors · 2025-11-20
> **Rebuttal by Authors (part 1)**
>
> Thanks for your review.
>
> **Clarifying the research objective in the context of GD:**\
> TL;DR: We don't study the stability of GD along its trajectory, only analyze the stability of fixed points (stability along the trajectory $ \neq $ convergence of GD)
>
> We believe there may be a misunderstanding of our paper, and we would like to clarify our research goal. Understanding the properties of the solutions that GD converges to can reveal important characteristics of the trained model. For example, linearly stable minima, which are potential limit points of GD, obey $ \\lambda\_{\\max} \leq 2/\\eta  $. From this condition, under the assumption that GD converges to linearly stable minimizers, we learn that when training a model with GD, the resulting input–output predictor function must be smooth (Nacson et al., 2023). Note that under this perspective, the full trajectory of the algorithm is less important. Only the ability of the algorithm to stay in the vicinity of a solution (local stability of a fixed point) is important. Our analysis therefore concentrates on the final stage of training, when the iterates are near a minimizer.
>
> Cohen et al. (2021) showed that the typical scenario for training NN is EoS. This motivates us to study the potential limit points of GD under EoS. The potential limit points of the dynamics correspond to period-2 cycles. If a minimizer admits a stable period-2 cycle and the iterates happen to arrive nearby, the iterates will converge to that cycle; if not, they will be repelled, and GD will not settle there. Overall, our work focuses on a local analysis near an equilibrium, aiming to identify the exact stability condition for period-2 cycles in high dimensions. This is precisely what Theorem 1 provides.
>
> Regarding your comment that “The result of Cohen et al. (2021) can be seen as a surprising statement that even thought the system is nonlinear, it acts as a linear one approximately”, we interpret this as referring to the observation that the global stability (non-divergence) of GD can be explained by local third-order Taylor approximations (as discussed in Damian et al., 2023). Although we assume EoS, this phenomenon is not related to our contribution.
>
>
> **Overlap between the paper’s GD contributions and prior work by Chen \& Bruna (2023):**\
> We thank the reviewer for bringing this paper to our attention. It provides an excellent opportunity to highlight the strength of our results and to clarify where prior work was unable to obtain them. Chen & Bruna (2023) analyzed the univariate case and derived the exact condition for stable oscillations in this case. However, it is clear that the extension of this condition to the multivariate setting was not known to them. In particular, for their multivariate analysis, they proposed applying the univariate condition along the sharpest direction of the Hessian. As we show in the comment above, and as follows from our multivariate result, this is not how the stability condition generalizes. The extension to higher dimensions is nontrivial.
>
> A detailed comparison and explanation are provided in our comment above. We will revise the paper to acknowledge that this setting was previously considered and to include the comparison, which makes our contribution over prior work explicit. Thank you again for the helpful pointer!
>
>
> **Previous work on the stability of SGD:**\
> Thank you for these references. In brief, Wu et al. (2023) study two notions of stability: linear stability and loss stability. Their loss stability result follows the same idea as Wu et al. (2022), which we already cite in the related work section. More specifically, they propose a necessary condition based on an alignment property for a particular regression setting (models with a one-dimensional output). However, a general analytic bound for this property is missing; there is no explicit lower bound on $\\mu\_{0}$, and it may even be zero.
>
> Their linear stability analysis assumes linearized dynamics, which we show do not capture the true nonlinear behavior of SGD. The same limitation applies to Andreyev \& Beneventano (2025), who also rely on quadratic approximations (i.e., linear dynamics). Moreover, as you noted, Andreyev \& Beneventano (2025) analyze stability along the trajectory, whereas our focus is on stability in the neighborhood of minimizers. This distinction has a tangible difference: their proposed measures are undefined in our setting of interpolating minima, since the denominators of both Batch Sharpness and Gradient–Noise Interaction vanish at the minimum.
>
> Nonetheless, we will include these references in the related work section. Thanks!

---

> ### Author Response · Authors · 2025-11-20
> **Rebuttal by Authors (part 2)**
>
> **Experiments and suboptimal results:**\
> We want to clarify again that we do not claim to have results showing EoS for the stochastic setting.
>
> The goal of our necessary condition on the stability of SGD (Theorem 2) is to highlight that prior work analyzes linearized dynamics without a valid theoretical foundation. A practical interpretation of this condition is that the maximal step size permitting stable SGD iterates near a minimizer is limited by the maximal sharpness of any batch that exhibits unstable oscillations (defies the condition in Theorem 1). While our intuition is that this may reflect the actual behavior of SGD, we currently do not have a method to verify it.

---

### Official Review · Reviewer_2TgM · 2025-11-01

**Soundness:** 2
**Presentation:** 3
**Contribution:** 3
**Rating:** 6
**Confidence:** 3

**Summary:**

The paper analyzes nonlinear local dynamics of gradient descent (GD) and stochastic GD (SGD) near minima, going beyond quadratic or linearized models. It shows that GD can remain stable through a period-2 cycle even when linear analysis predicts instability, with stability determined by higher-order derivatives (the Lyapunov coefficient of a flip bifurcation). For SGD, stability follows a “worst-case batch” rule: if any mini-batch would make GD diverge, full SGD diverges in expectation; if all batches are linearly stable, the dynamics contract near interpolating solutions. Simple toy numerical example is given to support the claims.

**Strengths:**

-	Understanding the training dynamics of gradient descent and stochastic gradient descent is an interesting research question.
-	The proposed theory explains period-2 cycle dynamics of GD beyond standard linear stability, which appears to be new.

**Weaknesses:**

-	The results focus on isolated minima, which differ from the many connected minima typically found in deep learning, as the authors note.
-	The SGD analysis assumes each batch has its own minimum and that batches are independent, which may be a somewhat strong assumption.

**Questions:**

-	Theorem 1 considers only the period-2 cycle. What about higher-period cycles, as shown in Figure 1(c)?
-	Theorem 1 assumes a step size of $\eta=2/\lambda$. Can additional results be derived for slightly larger $\eta$? Also, line 311 says “when $\eta$ is slightly above $\eta_c$”, what is the precise requirement on $\eta$?
-	Can the period-2 cycle statement from Kuznetsov, used in Section 5.1, be stated more precisely with full assumptions? The current version seems somewhat informal.

---

> ### Author Response · Authors · 2025-11-20
> **Rebuttal by Authors**
>
> Thanks for your review.
>
> **The proposed theory explains period-2 cycle dynamics of GD beyond standard linear stability, which appears to be new:**\
> Other reviewers brought to our attention prior work by Chen & Bruna (2022), which considered this setting. We provide a detailed comparison above. In brief, they present a rigorous analysis of the univariate case and derive the condition for stable oscillations. However, as we discuss in detail above, it was unclear to them how this condition generalizes to the multivariate setting, highlighting that our multivariate result is nontrivial.
>
>
> **Theorem 1 considers only the period-2 cycle. What about higher-period cycles, as shown in Figure 1(c)?**\
> Our analysis focuses on the typical case where the dynamics operate at the edge of stability, which corresponds to a period-2 cycle. Although higher-order cycles (such as period-4) can in principle occur, they appear to be less likely. At present, we do not have the mathematical tools to prove the existence of stable higher-period oscillations in high dimensions. Nevertheless, the condition in Theorem 1 would remain a necessary condition for any such oscillations.
>
> **Can the period-2 cycle statement from Kuznetsov, used in Section 5.1, be stated more precisely with full assumptions? The current version seems somewhat informal.**\
> We will work to clarify the assumptions in Section 5.1. The setting of our Theorem 1 exactly matches the setting in the book. Unfortunately, Kuznetsov did not present this result as a theorem in his book, so we cannot provide a direct quotation, and of course, we should not attempt to formulate one on his behalf. That said, we have outlined all the relevant assumptions and details (though perhaps not clearly enough). Fortunately, the book is available online [in this link]( https://www.ma.imperial.ac.uk/~dturaev/kuznetsov.pdf).
>
>
> **Theorem 1 assumes a step size of $ \\eta = 2/\\lambda $. Can additional results be derived for slightly larger $ \\eta $? Also, line 311 says “when $ \\eta $ is slightly above $ \\eta\_{c} $”, what is the precise requirement on $ \\eta $?**\
> In short, we do not know whether this analysis can be extended to learning rates slightly above the threshold. Our result relies on the analysis in Kuznetsov (1998), which uses center manifold theory to reduce the dynamics from high dimensions to a one-dimensional system, under the assumption that the non-dominant Hessian eigenvalues satisfy $ \\lambda\_i \\eta < 2 $. In this regime, the system has no unstable manifold, which guarantees correspondence between the reduced dynamics and the actual dynamics of GD in high dimensions.
>
> However, when $ \\eta > 2 / \\lambda\_{\\max} $, there is no center manifold; instead, an unstable manifold appears (hyperbolic fixed point). In this case, even if one performs a similar dimensionality-reduction step, the existence of a stable cycle is no longer guaranteed and requires additional assumptions. That said, such an analysis would not materially change the condition in Theorem 1, which would still act as a necessary, but not sufficient, condition for the existence of stable cycles.

---

### Official Review · Reviewer_SPrq · 2025-11-03

**Soundness:** 3
**Presentation:** 3
**Contribution:** 2
**Rating:** 4
**Confidence:** 3

**Summary:**

The paper studies nonlinear stability of GD/SGD near interpolating minimizers. The work gives an iff condition on second, third and fourth derivatives for there to exist stable period-2 oscillations. For SGD, it establishes a necessary and separately sufficient condition for stability, with the former being that if there is superexpontial divergence on one of the batches we get divergence, and the latter that if the step size is below $2/\lambda_{\max}$ for each of the batches. The work provides proofs for the above statements, and motivating examples.

**Strengths:**

The proofs are technical and seem to be correct (as far as I checked); good structure of the write-up, clearing up the intuition of the proof
Good examples illustrating the theorems.

**Weaknesses:**

I have some reservations concerning the contribution of the results of the paper. Since there are no empirical experiments, the theoretical contribution is the only contribution of the paper.

- Concerning GD, The fact that we can have stable (period-2) oscillations when we go beyond the stability threshold seems to be very much known in the literature. In particular, Damian et al. “Self-Stabilization…” (2022) exactly uses that mechanism to show the self-stabilization of GD; in particular, that’s exactly what Chen and Bruna “Beyond EoS…” (2022) analyze, see also Ma et al. “Beyond the Quadratic Approximation…” (2022) work with subquadratic functions and showing that it is indeed the case in the case of NNs. Most importantly, the general condition for this stability is outlined in Kuznetsov (which you do mention), which you specialize to the case of GD.
- SGD analysis. Concerning the necessary condition, although the exact statement doesn’t seem to appear anywhere, the fact that one has moment explosions when you have expansions with positive probability appear in the previous literature, see e.g. Kesten “Random Difference Equations…” (1973). The exact assumption you are using - the super-exponential growth, is a too strong of an assumption in such case, and would expectedly lead to first-moment explosion. Same goes for the sufficient condition. A similar condition appears in Diaconis and Freedman “Iterated Random Functions” (1998); and your strengthening of it to the uniform contraction of $\max_B\lambda_{\max}(\nabla^2L_B) < 2/\eta$ is seemingly too strong (I understand you are doing it for convenience), and in a sense expected to lead to stability. Therefore, these results seem to be known/expected - and, on the other hand, it is unclear whether these assumptions aren’t “too strong”.
- Which brings me to the question about the empirics. Considering that the results seem to be already known in the literature, and the exact assumptions are very strong, it would be important to see whether maybe those are applicable/useful in any real-world scenarios (and, in particular, in the case of NN). In particular, your condition is less “tight” than the sufficient condition of Wu et al. (2018) (because yours comes from the analysis of non-linear dynamocs) — and they show that their condition is not tight, so it is not clear whether this condition is applicable/useful. Mulayoff & Michaeli (2024) do show for example that their lower bound is close. Moreover, the empirical work of Andreyev & Beneventano “Edge of Stochastic…” (2025) in some of their experiments measure a statistic of $\lambda_{\max}(\nabla^2L_B)$ (expectation it is, which would lower bound your condition), and show that it is not a tight, and a comparison would be useful here.

**Questions:**

- What is your intuition about applicability of your conditions in real-world scenarios?
- Could you please write a more detailed comparison of your results to what has already been done in the literature?

---

> ### Author Response · Authors · 2025-11-20
> **Rebuttal by Authors (part 1)**
>
> Thanks for your review.
>
>
> **Comparison to prior work on the stability of oscillations in GD:**
> * Ahn et al. (2022) (cited) - This work focuses on loss fluctuations, introducing tools such as directional smoothness and the relative progress ratio to formalize this behavior. Using these metrics, the authors show that the stable behavior of GD arises from structured oscillations. However, although the paper provides explicit formulas for these quantities, it does not establish conditions under which these oscillations remain stable.
> * Ma et al. (2022) (cited) - This work analyzes edge-of-stability (EoS) behavior under the assumption that the loss exhibits subquadratic growth. The authors begin with the general univariate case and extend their analysis to higher dimensions under a specific structural assumption. Within this framework, GD with a fixed step size exhibits only two possible outcomes: it either enters a periodic cycle or converges to a minimum, no other behavior can occur. In other words, only a supercritical bifurcation is possible in their setting. This contrasts with our framework, in which we derive conditions that determine the type of bifurcation: supercritical (leading to stable oscillations) or subcritical (leading to unstable oscillations). Moreover, as noted by Chen & Bruna (2023), the subquadratic-growth assumption does not capture many practical ML scenarios, such as regression with the square loss (see discussion in Chen & Bruna (2023)).
> * Damian et al. (2023) (cited) - This work explains how GD self-stabilizes. In particular, it shows that when the iterates momentarily diverge along the sharpest eigenvector of the Hessian, they simultaneously move in the negative gradient direction of the curvature (i.e., effectively taking a gradient step with respect to the top eigenvalue of the Hessian). This motion leads to a reduction of the sharpness toward $ 2/\\eta $. A similar mechanism operates in our setting. However, taking a gradient step does not necessarily imply that the objective decreases (otherwise stability would be trivial) and additional conditions are required to ensure descent, including in the case of gradient steps on the sharpness. This work does not provide such conditions for the descent of the sharpness.
> * Chen \& Bruna (2023) (missing citation) - This work is perhaps the closest to ours, thank you for sharing it! The authors study the same setting we consider and begin with the univariate case, where they derive the condition for stable oscillations. However, as we discuss in detail above (see our separate comment), it was unclear to them how this condition extends to the multivariate setting. This underscores that our generalization is nontrivial and was not known to prior work. We will revise our paper to incorporate this work and include the discussion above.
>
> **Comparison to prior work on the stability of SGD:**\
> Prior work, such as Wu et al. (2018), Ma & Ying (2021), Wu et al. (2022), Lee & Jang (2023), Mulayoff & Michaeli (2024), and Andreyev & Beneventano (2025) (and others), relies on linearization, i.e. second-order Taylor approximation, to analyze the dynamics of SGD. In contrast, we study the nonlinear dynamics of SGD. While linearized dynamics is theoretically justified for stability analysis in the deterministic setting of GD, we show that there is no analogous justification for applying it to the stochastic setting in the way prior work has done. In particular, we provide analytic evidence that linearized dynamics fails to predict the behavior of the true nonlinear system.
>
> As is well understood from GD, once the step size exceeds the linear stability threshold, higher-order terms become essential. Some resulting oscillations are benign, whereas others are fatal, where the precise high-dimensional condition distinguishing these cases is given in Theorem 1. However, linearized SGD dynamics does not capture this distinction. When the step size exceeds the linear stability threshold for a given batch, and the batch loss does not satisfy the condition in Theorem 1, the iterates can blow up rapidly, causing the entire process to diverge.

---

> ### Author Response · Authors · 2025-11-20
> **Rebuttal by Authors (part 2)**
>
> **Applicability of the results to real-world scenarios:**\
> Our result for GD (Theorem 1) is exact and applies to any real-world scenario, assuming the minimum has a full-rank Hessian. This assumption covers common scenarios, for example, when $ \\ell\_{2} $ regularization is used. Another aspect of practical relevance is that our analysis shows that, for stability, the third derivative must dominate the fourth. This insight suggests a way to enhance the stability of GD in practice: one can introduce a cubic regularization term (potentially applied to the distance of the iterates to a slowly updated exponential moving average), which increases the dominance of third-order terms and thus improves stability. Note that doing the same regularization with $ \\ell\_{2} $ will impair stability. We leave this to future work.
>
> The goal of our necessary condition on the stability of SGD (Theorem 2) is to highlight that prior work analyzes linearized dynamics without a valid theoretical foundation. A practical interpretation of this condition is that the maximal step size permitting stable SGD iterates near a minimizer is limited by the maximal sharpness of any batch that exhibits unstable oscillations. While our intuition is that this may reflect the actual behavior of SGD, we currently do not have a method to verify it.
>
> We acknowledge that our sufficient condition for the stability of SGD (Theorem 3) is not tight for typical machine-learning tasks. However, it is optimal in an important sense, something we did not emphasize sufficiently in the paper. It is the strongest result one can derive under the following setting:
> * One aims to prove the stability of a nonlinear dynamical system,
> * The analysis is performed in expectation, and
> * No assumptions are imposed on third or higher-order derivatives.
>
> The simplest illustration of this optimality is Proposition 1, where our sufficient condition is also necessary. This perspective suggests a natural direction for future work. For instance, one could assume that every batch loss exhibits benign oscillations. Such assumptions may enable a more realistic analysis that leads to stability conditions more closely aligned with the true behavior of SGD.
>
> **Known/expected results**:\
> In comparing our work with Chen & Bruna (2023), we show that our GD result was not known to them and that Theorem 1 provides a nontrivial generalization. Our necessary condition for SGD also contradicts prior analyses that rely on linear stability, and we believe its publication would benefit the community. Especially since, as you noted, the exact statement does not appear elsewhere. Finally, while Kesten (1973), Diaconis & Freedman (1998), and Kuznetsov (1998) offer powerful mathematical tools, applying them in our context is itself novel.

---

### Author Response · Authors · 2025-11-20
**Detailed comparison to Chen & Bruna (2023) (comparison part 1)**

Chen \& Bruna (2023) analyze the dynamics of GD on a univariate function $ f: \\mathbb{R} \\to \\mathbb{R} $, where they derive the exact condition for stable oscillations, $ 3[f\^{(3)}]\^2 > f'' f\^{(4)} $ (Theorem 1 in their paper). Our Theorem 1 extends this condition to the multivariate setting. To highlight that this generalization is nontrivial and was not known in prior work, we compare our result with both the matrix factorization analysis and the experiments presented in Chen \& Bruna (2023).

Specifically, for matrix factorization, Chen \& Bruna (2023) used, without theoretical justification, the condition $ 3[f\_{\\Delta}\^{(3)}]\^2 > f\_{\\Delta}'' f\_{\\Delta}\^{(4)} $. Here $ f\_{\\Delta} $ denotes the loss restricted to the direction of the top eigenvector $ \\Delta $ of the Hessian (Theorem 6, Eq. 17). Moreover, in their experiments, they explicitly aim to demonstrate that this condition is *necessary*, i.e., that it must hold in practice (see Appendix B.3.2). This indicates that they genuinely believed  $ 3[f\_{\\Delta}\^{(3)}]\^2 > f\_{\\Delta}'' f\_{\\Delta}\^{(4)} $ to be the correct multivariate generalization of the univariate condition.

However, our results show that this is not the case. For instance, in the discussion below, we present a parametric function $ f\_{\\beta} $ for which this condition is never satisfied, yet the dynamics can exhibit either stable or unstable oscillations depending on the value of $ \\beta $. In addition, Chen & Bruna note that their condition sometimes fails to hold in their experiments, attributing these discrepancies to approximation error.

We see that the condition $ 3[f\_{\\Delta}\^{(3)}]\^2 > f\_{\\Delta}'' f\_{\\Delta}\^{(4)} $ does not correctly characterize the stability of the oscillations. Nevertheless, it remains a natural condition to examine. Can we understand what this condition actually means? Let $ \\{ ( \\lambda\_i, \\boldsymbol{v}_i ) \\}\_{i = 1}^{d} $ denote the eigenpairs of the Hessian at the minimum ($ \\Delta = \boldsymbol{v}\_{\\max} $). Then, it is easy to see that $ 3[f\_{\\Delta}\^{(3)}]\^2 > f\_{\\Delta}'' f\_{\\Delta}\^{(4)} $ can be written as
$$ \\tag{1}
3 \\frac{  \\left[  \\partial\^3\_{ \\boldsymbol{v}\_{\\max} } f   \\right] \^2  }{  \\lambda\_{\\max}  }> \partial\_{ \\boldsymbol{v}\_{\\max} }\^{4}   f,
$$
where $\partial\_{ \\boldsymbol{v} } $ is the directional derivative in the direction $  \\boldsymbol{v}  $, and $ f: \\mathbb{R}\^d \to \\mathbb{R}  $ is the multivariate funcion. In a comment below, we show that our exact condition can be written as
$$\\tag{2}
 3 \\sum\_{i=1}^d \\frac{ \\left[  \\partial\_{\\boldsymbol{v}\_i } \\partial\_{\\boldsymbol{v}\_{\\max} }\^2 f   \\right] \^2 }{ \\lambda\_i }> \partial\_{ \\boldsymbol{v}\_{\\max} }\^{4}   f.
$$
This alternative form of Theorem 1 shows how the univariate condition generalizes to the multivariate setting, in a non-trivial way. It also interprets the condition $ 3[f\_{\\Delta}\^{(3)}]\^2 > f\_{\\Delta}'' f\_{\\Delta}\^{(4)} $, which accounts for only a single term in the full series. Since all eigenvalues are positive, if (1) is satisfied, so is (2). Namely, our theory reveals that the stability of GD while restricted to the sharpest direction is a sufficient condition, and not necessary, as prior work believes. In particular, this sufficient condition is expected to be loose, as $\\left[ \\partial\^3\_{ \\boldsymbol{v}\_{\\max} } f \\right] \^2 $ is divided by $ \\lambda\_{\\max} $, while the exact condition in (2) contains a (long) series of fractions with smaller denominators.

---

> ### Author Response · Authors · 2025-11-20
> **Counterexample to Chen & Bruna (2023) (comparison part 2)**
>
> Let us consider the dynamics of GD on the two-dimensional function
> $$
> f ( \\boldsymbol{x}) = \\frac{1}{2}x\_1\^2 +\frac{1}{10}x\_2\^2 + \\beta x\_1\^2 x\_2   + \\frac{1}{10} x\_1\^4 ,
> $$
> in the vicinity of the minimum $ \\boldsymbol{x}\^{\*} = (x\_1\^{\*},x\_2\^{\*})  = (0,0) $. Here, the Hessian's eigenvalues at the minimum $ \\boldsymbol{x}\^* $ are $  \\lambda\_{\\max} = 1, \ \\lambda\_{\\min} = 0.2 $ and the corresponding eigenvectors are $ \\boldsymbol{v}\_{\\max} = (1,0)\^{\top} , \  \\boldsymbol{v}\_{\\min} = (0,1)\^{\top}$ .
> $$
> 3 \\frac{ \\left[  \\partial\_{ \\boldsymbol{v}\_{\\max} }\^3  f (0,0) \\right] \^2 }{ \\lambda\_{\\max} }  =
> 3 \\frac{ \\left[  \\partial\_{ x\_{1} }\^3  f (0,0) \\right]\^2 }{ \\lambda\_{\\max} }  = 0
> $$
> Additionally,
> $$
>  \\partial\_{ \\boldsymbol{v}\_{\\max} }\^4  f (0,0) =  \\partial\_{ {x\_1} }\^4  f (0,0) = \\frac{4!}{10} = 2.4 > 0 = 3 \\frac{ \\left[  \\partial\_{ \\boldsymbol{v}\_{\\max} }\^3  f (0,0) \\right] \^2 }{ \\lambda\_{\\max} } .
> $$
> We see that the condition proposed by Chen \& Bruna (2023), *i.e.* $ 3[f\_{\\Delta}\^{(3)}]\^2 > f\_{\\Delta}'' f\_{\\Delta}\^{(4)} $, does not hold for any value of $ \\beta $. However,
> $$
> 3 \\frac{ \\left[ \\partial\_{ \\boldsymbol{v}\_{\\min} } \\partial\_{ \\boldsymbol{v}\_{\\max} }\^2  f (0,0) \\right] \^2 }{ \\lambda\_{\\min} }  =
> 3 \\frac{ \\left[ \\partial\_{ x\_{2} } \\partial\_{ x\_{1} }\^2  f (0,0) \\right] \^2 }{ \\lambda\_{\\min} }  =
> 3 \\frac{ \\left( 2 \\beta \\right)\^2} { 0.2 } = 60 \\beta\^2.
> $$
> Our exact condition for stable oscillations,
> $$
>  3 \\sum\_{i=1}^2 \\frac{ \\left[  \\partial\_{\\boldsymbol{v}\_i } \\partial\_{\\boldsymbol{v}\_{\\max} }\^2 f   \\right] \^2 }{ \\lambda\_i }> \partial\_{ \\boldsymbol{v}\_{\\max} }\^{4}   f,
> $$
> is $ 60 \\beta\^2 > 2.4 $, which reduces to $  | \\beta | > 0.2 $. Then, our theory asserts that whenever $  | \\beta | > 0.2 $, the oscillations are stable; otherwise, the oscillations are unstable. You can find a simulation of this behavior in the paper, on the last page of the appendix. When we run GD while scanning on $ \\beta $, we get the exact phase transition predicted here.

---

> ### Author Response · Authors · 2025-11-20
> **Equivalent form of Theorem 1 (comparison part 3)**
>
> Our condition for stable oscillations in Theorem 1 is
> $$
> \\mathcal{D}\^{3} \\mathcal{L}( \\boldsymbol{x}\^* ) [\\boldsymbol{v}\_{\\max},\\boldsymbol{v}\_{\\max}, \\boldsymbol{q}] > \\mathcal{D}\^{4} \\mathcal{L}( \\boldsymbol{x}\^* ) [\\boldsymbol{v}\_{\\max},\\boldsymbol{v}\_{\\max}, \\boldsymbol{v}\_{\\max}, \\boldsymbol{v}\_{\\max}]
> $$
> where
> $$
> \\boldsymbol{q} \\triangleq \\Big[ \\nabla^2 \\mathcal{L}( \\boldsymbol{x}\^*  )  \\Big]^{-1} \\nabla\_{\\boldsymbol{v}} \\mathcal{D}\^{3} \\mathcal{L}( \\boldsymbol{x}\^* ) [\\boldsymbol{v},\\boldsymbol{v}, \\boldsymbol{v}] \Big|\_{\\boldsymbol{v} = \\boldsymbol{v}\_{\\max}}.
> $$
> As we noted in the paper (line 200), $  \\nabla\_{\\boldsymbol{v}} \\mathcal{D}\^{3} \\mathcal{L}( \\boldsymbol{x}\^* ) [\\boldsymbol{v},\\boldsymbol{v}, \\boldsymbol{v}] \Big|\_{\\boldsymbol{v} = \\boldsymbol{v}\_{\\max}} =  3 \\mathcal{D}\^{3} \\mathcal{L}( \\boldsymbol{x}\^* ) [\\boldsymbol{v}\_{\\max},\\boldsymbol{v}\_{\\max}] $. Let $ \\{ ( \\lambda\_i, \\boldsymbol{v}_i ) \\}\_{i = 1}^{d} $ denote the eigenpairs of the Hessian at the minimum $ \\boldsymbol{x}\^* $, then
> $$
> \\Big[ \\nabla^2 \\mathcal{L}( \\boldsymbol{x}\^*  )  \\Big]^{-1} =  \\sum\_{i = 1}^d \\frac{1}{\\lambda\_{i} } \\boldsymbol{v}\_{i}  \\boldsymbol{v}\_{i}^{\top}.
> $$
> Thus,
> $$
> \\boldsymbol{q}  = 3 \\sum\_{i = 1}^d \\frac{1}{\\lambda\_{i} } \\boldsymbol{v}\_{i}  \\boldsymbol{v}\_{i}^{\top}\\mathcal{D}\^{3} \\mathcal{L}( \\boldsymbol{x}\^* ) [\\boldsymbol{v}\_{\\max},\\boldsymbol{v}\_{\\max}]  = 3 \\sum\_{i = 1}^d \\frac{\\mathcal{D}\^{3} \\mathcal{L}( \\boldsymbol{x}\^* ) [\\boldsymbol{v}\_{\\max},\\boldsymbol{v}\_{\\max}, \\boldsymbol{v}\_{i}] }{\\lambda\_{i} } \\boldsymbol{v}\_{i}    .
> $$
> Therefore,
> $$
> \\mathcal{D}\^{3} \\mathcal{L}( \\boldsymbol{x}\^* ) [\\boldsymbol{v}\_{\\max},\\boldsymbol{v}\_{\\max}, \\boldsymbol{q}] =
> \\mathcal{D}\^{3} \\mathcal{L}( \\boldsymbol{x}\^* ) \\left[\\boldsymbol{v}\_{\\max},\\boldsymbol{v}\_{\\max},
> 3 \\sum\_{i = 1}^d \\frac{\\mathcal{D}\^{3} \\mathcal{L}( \\boldsymbol{x}\^* ) [\\boldsymbol{v}\_{\\max},\\boldsymbol{v}\_{\\max}, \\boldsymbol{v}\_{i}] }  {\\lambda\_{i} } \\boldsymbol{v}\_{i} \\right] = 3 \\sum\_{i = 1}^d  \frac{ \\left( \\mathcal{D}\^{3} \\mathcal{L}( \\boldsymbol{x}\^* ) [\\boldsymbol{v}\_{\\max},\\boldsymbol{v}\_{\\max}, \\boldsymbol{v}\_{i}] \\right)^2 }  {\\lambda\_{i} } ,
> $$
> where the last step is by linearity. Overall, Theorem 1 is equivalent to
> $$
>  3 \\sum\_{i=1}^d \\frac{ \\left[  \\partial\_{\\boldsymbol{v}\_i } \\partial\_{\\boldsymbol{v}\_{\\max} }\^2 \\mathcal{L}  \\right] \^2 }{ \\lambda\_i }> \partial\_{ \\boldsymbol{v}\_{\\max} }\^{4}   \\mathcal{L} ,
> $$
> where $\partial\_{ \\boldsymbol{v} } $ is the directional derivative in the direction $  \\boldsymbol{v}  $.

---

### Author Response · Authors · 2025-12-03
**Summary for the Area Chair (AC)**

Dear AC,

We provide here a summary of the reviews. The primary concern, raised mainly by reviewer uH3Y and partly by reviewer SPrq, is that our results are already known or follow trivially from prior work. In our rebuttal, we refuted these claims. The reviewers specifically cited Chen & Bruna (2023) as having already addressed our problem. However, as we explained, their analysis covers only the univariate setting. For higher dimensions, they propose a heuristic extension based on a cross-section of the multivariate function. In a comment below, we show that this heuristic is incorrect by providing an explicit counterexample where their prediction fails. By contrast, our paper derives the full multivariate result, which matches the observed behavior of gradient descent. This constitutes a non-trivial and genuinely novel generalization. In this sense, our work corrects and clarifies the existing literature.

We also believe that reviewer uH3Y may have misunderstood key aspects of our paper, which likely contributed to the unreasonably low rating that he gave us (0). In our response to this reviewer, we clarified both our setting and our contributions. Unfortunately, although we posted our rebuttal a week before the discussion period was interrupted, none of the reviewers responded.

We kindly ask that you consider these points, as well as the overall positive balance of the reviews, when making your decision.

---

### Meta-Review · Area_Chair_uRo4 · 2026-01-04

**Summary:**

The paper examines the stability of gradient-based training beyond linearization. It shows that for deterministic GD, linear stability analysis can be misleading: nonlinear terms may induce stable oscillations near a linearly unstable minimum, with convergence occurring once the step size decays, and an exact condition involving higher-order derivatives is derived. For SGD, the authors argue that nonlinear dynamics can diverge in expectation if even a single minibatch is unstable, implying worst-case rather than average-case stability, while also proving stability in expectation when all batches are linearly stable.

While technically sound, the contribution is limited in novelty and scope. The central message, that nonlinear effects can invalidate linear stability analysis, is well known in dynamical systems and optimization, and the specific results appear to rely on restrictive local conditions with unclear generality or relevance in high-dimensional, practical settings. The SGD analysis adopts a worst-case batch perspective that is weakly connected to modern training practices, and the paper lacks empirical validation to substantiate the claimed phenomena or their implications for flat minima and generalization.

**Reviewer Concerns:**

Nothing to note.

**Reviewer Scores:**

Can't predict.

---

### Decision · Program_Chairs · 2026-01-26

Reject